# Hazardous thunderstorm intensification over Lake Victoria

Wim Thiery[1,2], Edouard L. Davin[2], Sonia I. Seneviratne[2], Kristopher Bedka[3], Stef Lhermitte[1,4] & Nicole P.M. van Lipzig[1]

Weather extremes have harmful impacts on communities around Lake Victoria, where thousands of fishermen die every year because of intense night-time thunderstorms. Yet how these thunderstorms will evolve in a future warmer climate is still unknown. Here we show that Lake Victoria is projected to be a hotspot of future extreme precipitation intensification by using new satellite-based observations, a high-resolution climate projection for the African Great Lakes and coarser-scale ensemble projections. Land precipitation on the previous day exerts a control on night-time occurrence of extremes on the lake by enhancing atmospheric convergence (74%) and moisture availability (26%). The future increase in extremes over Lake Victoria is about twice as large relative to surrounding land under a high-emission scenario, as only over-lake moisture advection is high enough to sustain Clausius–Clapeyron scaling. Our results highlight a major hazard associated with climate change over East Africa and underline the need for high-resolution projections to assess local climate change.

[1] KU Leuven, Department of Earth and Environmental Sciences, Celestijnenlaan 200E, 3001 Leuven, Belgium. [2] ETH Zurich, Institute for Atmospheric and Climate Science, Universitaetsstrasse 16, 8092 Zurich, Switzerland. [3] NASA Langley Research Center, Science Directorate, 21 Langley Boulevard, Hampton, Virginia 23681, USA. [4] Delft University of Technology, Department of Geoscience and Remote Sensing, Stevinweg 1, 2600 GA Delft, The Netherlands. Correspondence and requests for materials should be addressed to W.T. (email: wim.thiery@env.ethz.ch).

Severe thunderstorms and associated high waves represent a constant threat to the 200,000 fishermen operating on Lake Victoria[1,2]. The International Red Cross assumes that 3,000–5,000 fishermen die every year on the lake[2], by which it substantially contributes to the global death toll from natural disasters. Each perished fisherman leaves on average eight relatives without an income, underlining the vulnerability of East African fishing communities to these natural hazards[1–3]. Despite the long-known bad reputation of Lake Victoria[4], the understanding of the drivers of these extreme thunderstorms remains limited[5]. Moreover, anthropogenic climate change may significantly affect these hazardous weather systems. In many parts of the world, future climate simulations project an intensification of precipitation extremes and associated weather conditions[6–11], but the potential future changes in extremes over Lake Victoria are still unknown.

In this study, we use a unique combination of state-of-the-art satellite remote sensing, a high-resolution regional climate model and coarser-scale ensemble simulations to project changes in extreme precipitation over Lake Victoria. We project a strong and robust increase in precipitation extremes over Lake Victoria and show that this increase is about double over the lake compared with surrounding land. Although the occurrence of extreme precipitation in the present-day climate is mostly controlled by atmospheric dynamics, its future intensification can be entirely attributed to the advection of more humid air over the lake.

## Results

**Satellite data analysis**. Satellite observations enable the recognition of severe weather by detecting overshooting tops (OTs), that is, dome-like protrusions atop a cumulonimbus anvil induced by intense updrafts[12,13]. OTs mark the presence of vigorous thunderstorms and are tightly linked to severe weather reports[12–15]. By applying an OT detection algorithm to Meteosat Second Generation observations (Methods), we establish a new severe thunderstorm climatology for East Africa. The results reveal a marked imprint of Lake Victoria on the diurnal thunderstorm cycle and confirm its status as one of the most convectively active regions on Earth[5,13,16–18] (Fig. 1). From 2005 to 2013, 73% of all 1,400,000 OT pixels detected over the lake occurred at night (22:00 to 9:00 UTC), in contrast to the surrounding land where afternoon storms dominate (72% of all 4,200,000 OT pixels during 9:00 to 16:00 UTC). Local evaporation and mesoscale circulation have been identified as key drivers of

the present-day diurnal cycle of precipitation over Lake Victoria[5,17–21], but so far it is not known how mean and extreme precipitation over this lake respond to a temperature increase induced by anthropogenic greenhouse gas emissions. To address this question, we performed a high resolution (~7 km grid spacing), coupled lake–land–atmosphere climate projection for the African Great Lakes region with the regional climate model COSMO-CLM[2], and analysed coarser-scale ensemble projections from the Coordinated Regional climate Downscaling Experiment (CORDEX) for the end of the century under a high-emission scenario (RCP8.5; Methods, Supplementary Fig. 1 and Supplementary Table 1).

**Extreme precipitation projections**. The projections show a contrasting change of mean and extreme precipitation over Lake Victoria (Fig. 2; Supplementary Fig. 2), with mean precipitation decreasing while the intensity of extreme precipitation increases. Moreover, by the end of the century the increase in extremes (precipitation above the 99th percentile) is 2.4 ± 0.1 times higher over the lake than over its surrounding land in the high-resolution projection (1.8 ± 1.0 times in the CORDEX ensemble). Today convection initiates in the eastern part of the lake and intensifies while being advected westwards along the trade winds[4,5]. In the future, storms are projected to release extreme precipitation more in the eastern part of the lake, leading to an eastward shift of intense precipitation (Fig. 2a,b).

In contrast to the increase in extremes, the annual mean precipitation projected by the high-resolution model declines over the lake by 6% (Supplementary Fig. 2 and Supplementary Note 1)[22]. This is also evident from changes in daily binned precipitation over the lake, which show an overall future drying except for precipitation above the 90th percentile (Fig. 2c,d). If we correct for this average drying (Methods), the effect of Lake Victoria on future extremes is even more pronounced, with the increase being 3.2 ± 0.3 times larger over Lake Victoria compared with surrounding land in COSMO-CLM[2] and even 4.2 ± 1.6 times larger in the CORDEX ensemble.

In other words, very intense storms are projected to become more frequent in the future over Lake Victoria. For example, by the end of the century a 1-in-15-year precipitation event over Lake Victoria becomes a 1-in-1.5-year event in the high-resolution projection (1-in-0.8-year event in the CORDEX ensemble). In both cases this exceeds the projected increase in storm frequency over land (Supplementary Fig. 3).

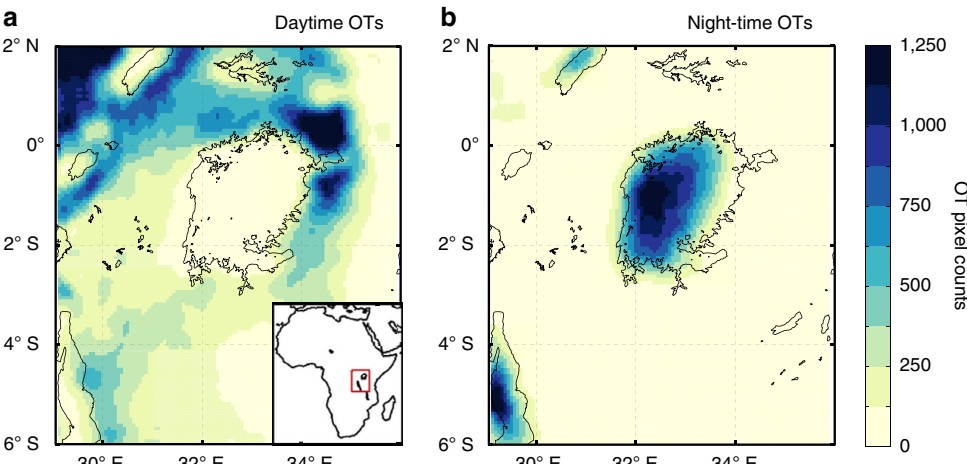

**Figure 1 | Lake imprint on severe thunderstorm occurrence.** (**a**,**b**) Satellite-based overshooting tops (OT) detections during 2005–2013 over the Lake Victoria region (red square in the inset panel), from 9:00 to 15:00 UTC and from 00:00 to 9:00 UTC, respectively, as derived from the Spinning Enhanced Visible and Infrared Imager (SEVIRI; Methods).

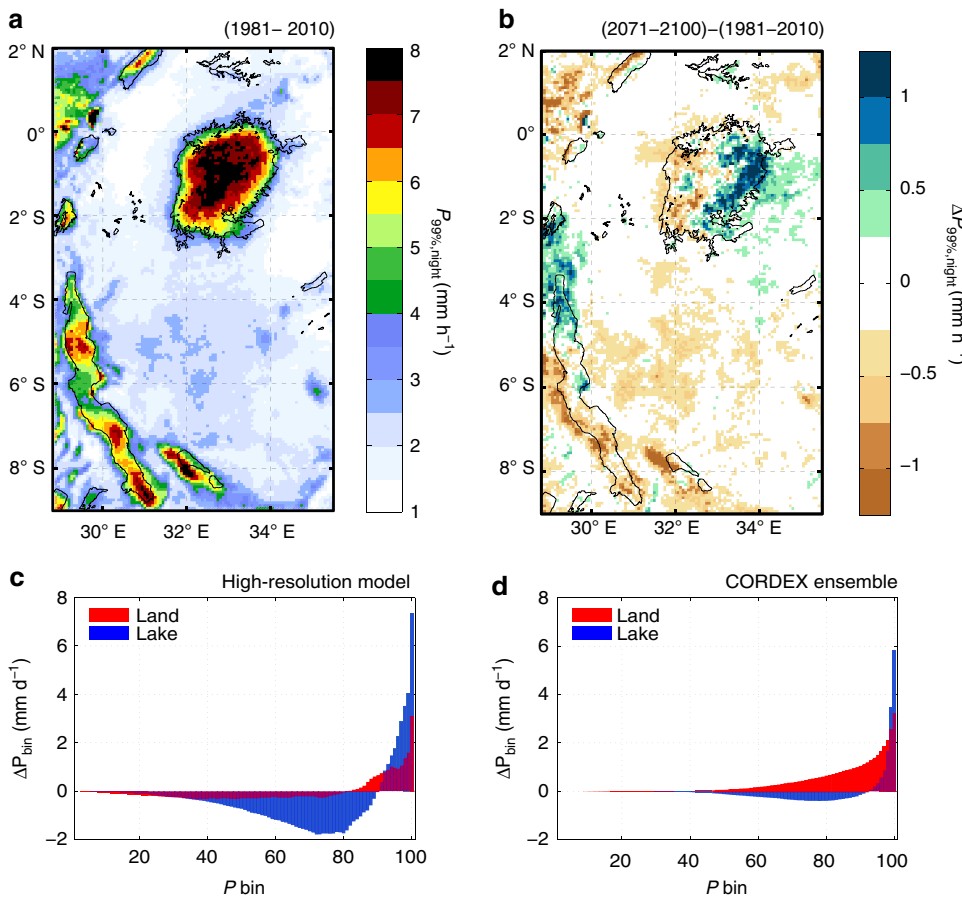

**Figure 2 | Projected end-of-century changes in extreme precipitation over Lake Victoria.** (**a**) Night-time 99th percentile precipitation ($P_{99\%,night}$, 00:00 to 9:00 UTC) and (**b**) its projected future change from the high-resolution COSMO-CLM² model. (**c,d**) 24 h Lake (blue bars) and surrounding land (red bars) binned precipitation change (P bin) from COSMO-CLM² and the ensemble mean of nine CORDEX-Africa members, respectively. The red rectangle in Supplementary Fig. 1 includes the land pixels considered as surrounding land. All changes are between time periods 1981–2010 and 2071–2100 under RCP8.5.

**Assessing uncertainty.** Based on a single high-resolution projection (~7 km), we cannot assess modelling uncertainties or compare emission scenarios. Since this type of simulations are computationally very expensive, this is a recurrent limitation of studies investigating climate change at high resolution[23–26]. By providing ensemble projections at coarser resolution (~50 km), the CORDEX initiative enables uncertainty assessments within the constraints of the quality of both the downscaling tool and the lateral boundary conditions[27]. Although some differences occur between the high- and coarse-resolution projections, it is clear that the lake effect on the future precipitation distribution is robust (Fig. 2c,d; Supplementary Fig. 3). This is further confirmed by the fact that the projected response in the coarse-resolution ensemble (Fig. 2d) is to a large extent independent of the driving global model. In particular, every CORDEX simulation projects a reduction in over-lake precipitation for all bins below the 90th percentile and an amplification of the increase in the highest bins, thereby corroborating the high-resolution model (Fig. 2). Comparison of the coarse-resolution RCP8.5 and RCP4.5 ensembles moreover demonstrates that the choice of the emission scenario does not influence Lake Victoria's amplifying role on extreme precipitation changes.

At the same time COSMO-CLM² clearly outperforms all CORDEX models as well as a state-of-the-art reanalysis in terms of precipitation representation, underlining the benefits of enhanced resolution and use of a lake model for climate simulations over the region (Supplementary Figs 4–6 and Supplementary Note 2)[5,28–31]. Decreasing the horizontal grid spacing to convection-permitting scales (below ~4 km) would most likely improve the skill of our climate simulations even more, since the convection parameterization employed in the high-resolution model still entails a number of limitations[25,26,32–35]. Overall these findings highlight the need for running coordinated high-resolution projections to quantify local climate change in regions with a particular dynamical regime[23].

**Driving mechanisms.** To better understand the processes controlling present-day extreme precipitation occurrence and its future change, we analysed observations and a multi-year reanalysis downscaling with COSMO-CLM² (Methods). Satellite observations of OTs and precipitation reveal that increased night-time thunderstorm activity and rainfall amounts over Lake Victoria are preceded by intense storms and rainfall over land the prior afternoon (Fig. 3a,b). Large-scale moisture availability contributes to this positive relationship, but alone it cannot explain the observed correlation (Supplementary Note 3). Land storms therefore act as a positive feedback for the intensity of night-time lake storms. These severe land storms could impact storm intensity over the lake in two ways. First, they could enhance moisture convergence by increasing the near-surface-specific humidity (thermodynamic control; Fig. 3c, Supplementary Fig. 7). Second, they could modify the lake/land breeze system[5] by cooling the land surface (dynamic control).

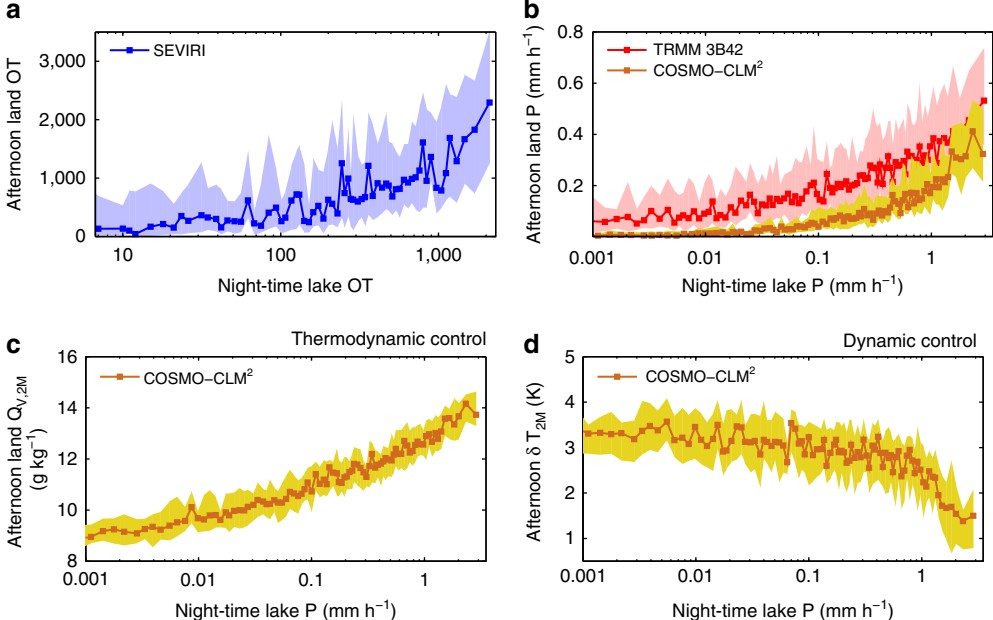

**Figure 3 | Afternoon controls on night-time extreme precipitation.** (**a**) Afternoon SEVIRI overshooting tops (OT) pixel detections over land surrounding Lake Victoria versus night-time OT pixels over the lake (2005–2013; blue). (**b**) Afternoon TRMM 3B42 precipitation (P) around Lake Victoria versus precipitation over the lake (1998–2013; red) and corresponding modelled values from a 10-year reanalysis downscaling with COSMO-CLM$^2$ (1999–2008; brown) (Methods). (**c,d**) Same as **b**, but for the afternoon land 2-m specific humidity ($Q_{V,2M}$) and lake–land temperature contrast ($\delta T_{2M} = T_{2M,land} - T_{2M,lake}$), respectively, as derived from the reanalysis downscaling. Each variable on the y axis was binned according to the variable on the x axis using a bin width of 1%. Full lines indicate the bin median and shaded uncertainty bands the interquartile range. Note the logarithmic x axis.

In that case the cold pools of the afternoon storms act to reduce gradients in near-surface air temperature between lake and land (Fig. 3d), thereby weakening the lake breeze and possibly also moisture transport away from the lake. If the cold anomaly persists into the night, this could strengthen the land breeze and by that possibly stimulate moisture convergence and column instability[36]. Interestingly, lake evaporation does not control the occurrence of extremes over Lake Victoria, despite its key role in the regional hydrological cycle[5,17,19].

Given the importance of moisture convergence for triggering precipitation extremes over Lake Victoria, we investigate whether dynamic or thermodynamic controls on moisture convergence dominate and how this might change towards the future (Methods). In the present-day climate, moisture convergence more than triples during 24 h periods (9:00 to 9:00 UTC) with extreme night-time precipitation compared with average conditions ($81 \times 10^{10}$ versus $26 \times 10^{10}$ kg d$^{-1}$ on average). A large fraction (74%) of this increase can be attributed to dynamical effects, while only 26% is due to the enhanced moisture content of converging air masses (Supplementary Fig. 7 and Supplementary Table 2). We thus conclude that mesoscale circulation is crucial for triggering extremes in the present-day climate (see also Supplementary Notes 3 and 4).

For the end-of-the-century projection, in contrast, we find that the intensification of precipitation extremes is entirely due to the enhanced moisture content of converging air masses. Under RCP8.5, the model projects a 27% increase in moisture convergence during extremes. This rise is entirely attributed to thermodynamic effects as dynamical changes reduce moisture convergence by 3% (Supplementary Table 2). The increase in moisture convergence is consistent with the modelled sensitivity of strong precipitation extremes (99.9th percentile) to temperature changes: only over the lakes the theoretically expected Clausius–Clapeyron scaling is attained, whereas over the surrounding land the scaling is constrained by moisture

availability (Supplementary Fig. 8 and Supplementary Note 5). Finally, we find no role for lake evaporation changes, as its increase during extremes is 50 times smaller than the rise in moisture convergence.

The picture is different for the decrease in annual mean precipitation, where mesoscale dynamical changes dominate. By the end of the century, night-time near-surface air temperatures will increase more rapidly over land compared with the lake, thus weakening the lake–land temperature contrast responsible for the land breeze, night-time moisture advection and updrafts. In addition, during daytime the warmer land will intensify the lake breeze and associated moisture divergence from the lake.

In summary, we have shown that new satellite-based detections of severe storms reveal a clear diurnal variation in storm activity over Lake Victoria and that nights with more intense storm activity are preceded by afternoons with more intense storms over the neighbouring land. Using a dedicated, high-resolution climate model set-up for equatorial East Africa, we found that these intense land storms favour moisture convergence by enhancing moisture availability but especially by weakening the afternoon lake breeze and strengthening the night-time land breeze (Fig. 4). We project a substantial future decline in annual mean precipitation over Lake Victoria, which may be explained by changing mesoscale dynamics associated with a faster warming land (Fig. 4). However, despite this average decrease, we project a strong and robust increase in precipitation extremes over Lake Victoria and show that this increase is about double over the lake compared with surrounding land. The rise in precipitation extremes is entirely due to enhanced future moisture availability (Fig. 4), and only over the lake the advection of more humid air supplies enough moisture to sustain Clausius–Clapeyron scaling. The increase in extremes is therefore not physically incongruous with the decrease in mean precipitation caused by mesoscale dynamical changes.

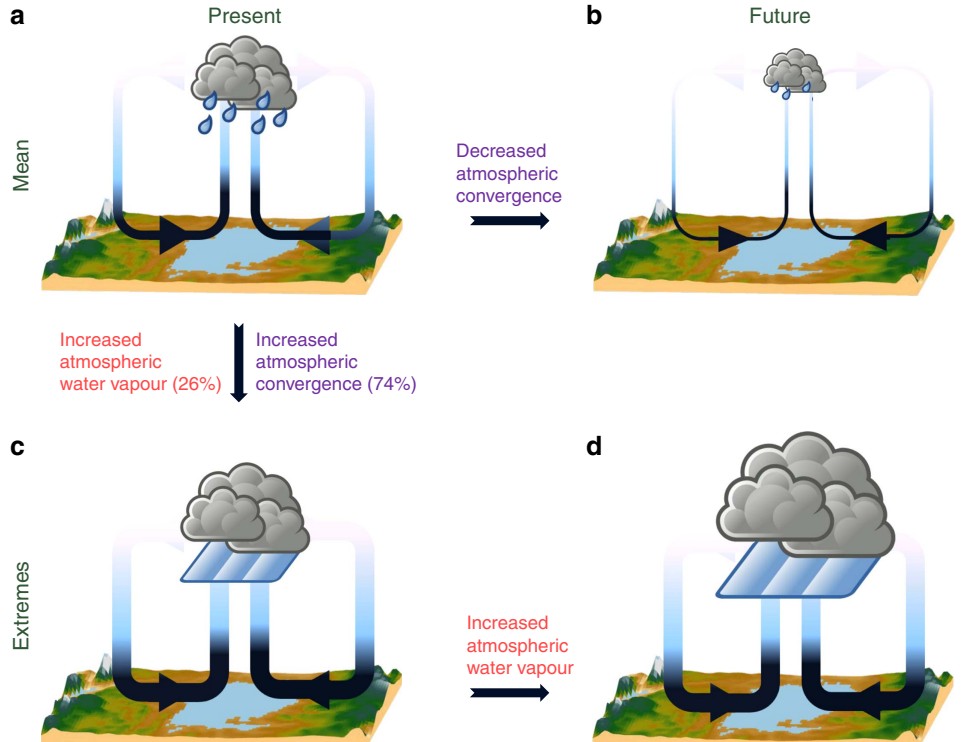

**Figure 4 | Processes controlling night-time precipitation extremes and climate change over Lake Victoria.** (**a**) In the present-day climate, local evaporation and net moisture flux convergence (MFC; Methods) along the land breeze both contribute to night-time precipitation generation over Lake Victoria. (**b**) Climate change simulations project a decrease in average precipitation, despite enhanced lake evaporation. Future mesoscale circulation changes impeding thunderstorm development are responsible for this decrease. (**c**) Present-day precipitation extremes are associated with increased MFC, of which 74% is explained by enhanced atmospheric convergence and the remaining 26% by enhanced moisture content of advected air masses (Fig. 3; Supplementary Fig. 7). (**d**) The future intensification of precipitation extremes is amplified over Lake Victoria compared with surrounding land and entirely due to higher moisture content of converging air masses.

## Discussion

Our results emphasize a major hazard associated with climate change over East Africa with potential severe human impacts. Lake Victoria directly sustains the livelihood of 30 million people living at its coasts and its fishing industry is a leading natural resource for East African communities[1,2]. However, given the projected increase in extreme over-lake thunderstorms, the current vulnerability of local fishing communities[2,3] and their growing exposure driven by rapid urbanization along the lakefront[37], this lake is likely to remain the most dangerous stretch of water in the world. At the same time, our findings mark an opportunity for developing a satellite-based early warning system for hazardous thunderstorms over Lake Victoria. Warning systems deriving predictions from the strong afternoon controls on night-time thunderstorms (Fig. 3) have the potential to substantially reduce the vulnerability of local fishing communities. This would complement ongoing efforts, in particular by the UK Met Office[18], to provide storm warnings for the region based on numerical weather prediction.

This study finally underscores the need for high-resolution projections to assess local climate change, especially in regions with a particular dynamical regime where extreme precipitation responses to anthropogenic climate change may be very different from large-scale projections[11,38,39]. High-resolution projections accounting for lake–atmosphere interactions are still very rare and may face challenges[27,40], but adopting this approach is critical to assess future climate impacts in regions where lakes are abundant.

## Methods

**Overshooting top detections.** We applied an overshooting top detection algorithm (OTDA)[12,14] to the Meteosat Second Generation (MSG) Spinning Enhanced Visible and Infrared Imager (SEVIRI) infrared satellite data for equatorial East Africa (23° E to 43° E; 11° S to 7° N). The SEVIRI instrument provides images at 15-min temporal and $\sim$4-km spatial resolution over the Lake Victoria region[41]. The OTDA builds on the premise that OTs are composed of a small region of very cold infrared brightness temperatures surrounded by a warmer cirrus anvil cloud[12,14,15]. As OTs penetrate through the level of neutral buoyancy (LNB), they continue to cool at a rate of 7–9 K km$^{-1}$ making them much colder than the anvil cloud which typically resides between the LNB and the tropopause[42]. The OTDA first identifies candidate OT regions by selecting SEVIRI pixels with an infrared brightness temperatures $\leq 217.5$ K and near to or colder than the tropopause temperature defined by the Modern Era Retrospective analysis for Research and Applications (MERRA). Subsequently the cirrus anvil cloud surrounding a candidate OT region is sampled, and if the candidate is substantially ($\geq 6$ K) colder than the anvil it is classified as an OT. Detection thresholds for the OTDA were based on the analysis of a large sample of OT-producing storms depicted within 1-km spatial resolution Moderate-resolution Imaging Spectroradiometer (MODIS) and Advanced Very High Resolution Radiometer (AVHRR) imagery in combination with OT product user feedback from the National Oceanic and Atmospheric Administration (NOAA) operational weather forecasting community. The OTDA finally corrects for parallax errors when locating OT-producing storms, thereby assuming a cloud top height of 16 km. Using this approach, more than 50 million OT pixels were detected from 2005 to 2013 over equatorial East Africa. A single OT is on average composed of 11 OT pixels and typically does not exceed 15 km in diameter.

**Climate simulations.** All simulations were performed with COSMO-CLM$^2$, which couples the non-hydrostatic regional climate model COSMO-CLM version 4.8 to the Community Land Model version 3.5 (CLM3.5) and the Freshwater Lake model (FLake)[43,44]. Detailed descriptions of this state-of-the-art model system and its subcomponents are provided in earlier studies[5,43–49].

The COSMO-CLM$^2$ model was applied in its tropical configuration[5,47] to generate three climate simulations. First, a control simulation (CTL) was conducted

with the ERA-Interim reanalysis as lateral boundary conditions for the period 1996–2008 and using the 0.44° COSMO-CLM CORDEX-Africa evaluation simulation[47] as intermediate nesting step. The same nesting strategy was employed to dynamically downscale a global climate model (GCM) simulation from the Coupled Model Intercomparison Project phase 5 (CMIP5). The Max Plank institute MPI-ESM-LR GCM was selected based on its high skill over East Africa[50]. GCM downscalings were performed for the historical reference period 1978–2010 (HIS) and the future projection 2068–2100 under the high emission scenario RCP8.5 (FUT). This scenario was chosen as it is expected to facilitate interpretation by yielding a strong climate response and as it provides the likely upper bound of the changes which may be expected by the end of the century.

All experiments were conducted at a horizontal resolution of 0.0625° (∼7 km), using 50 vertical levels and a time step of 60 s (Supplementary Table 1). Moist convection was parameterized by the Tiedtke mass flux scheme[48]. The model domain encompasses the central part of the East African rift (Supplementary Fig. 1) and therewith includes most of the African Great Lakes. The first 3 years of each simulation were considered as spin-up and excluded from the analysis. Overall, the simulations are designed to simulate the influence of a high-emission scenario on mean and extreme precipitation over and around Lake Victoria. Large-scale precipitation changes (for example, over the whole of East Africa[40]) and influences of decadal variability[51,52] are thereby beyond the scope of this study.

**Data binning and correction for average drying.** Binned precipitation changes $\Delta P_{bin}$ (Fig. 2c,d) were computed using a 1% bin width and assuming tied ranks. The lake influence on extreme precipitation changes was computed as the ratio between the mean daily precipitation change over lake and land for the highest bin (containing precipitation above the 99th percentile). Uncertainty ranges were derived as the maximum difference between this ratio and the ratio obtained with one standard deviation added or subtracted from the mean, respectively. In addition, the binned change was also corrected for the change in mean precipitation. The correction is performed by subtracting from each precipitation bin change $\Delta P_{bin}$ the fractional contribution to the average precipitation change assuming equal weights ($P_{bin}\Delta P/P$). By doing so the integral over all bins becomes zero and only the lake influence on the precipitation distribution is retained.

**Moisture convergence calculation.** The vertically integrated, instantaneous moisture flux convergence (MFC, kg s$^{-1}$) over Lake Victoria was computed following

$$\text{MFC} = \oint \int_{P}^{P_0} q u_n \frac{dp}{g} dc \qquad (1)$$

along the red circle denoted in Supplementary Fig. 1. The specific humidity is indicated by $q$ (kg kg$^{-1}$), $u_n$ is the wind velocity (m s$^{-1}$) normal to the contour's outer edge (outward defined positive), $g$ the standard gravitational acceleration and dp the segmented pressure differences (Pa) between the surface pressure $P_0$ and the pressure $P$ taken at a height of 7 km above sea level. The contour segments $dc$ (m) are defined using the integer midpoint circle algorithm with 1.4° radius and centred at 1°S–33° E (Supplementary Fig. 1). Given the total change in moisture convergence during extremes $\Delta\text{MFC}_{tot} = \text{MFC}_{EX} - \text{MFC}_{CTL}$, and the change induced by atmospheric dynamics given by

$$\Delta\text{MFC}_{dyn} = \oint \int_{P}^{P_0} q_{EX} \frac{\overline{q_{CTL}}}{\overline{q_{EX}}} u_{n,EX} \frac{dp}{g} dc - \oint \int_{P}^{P_0} q_{CTL} u_{n,CTL} \frac{dp}{g} dc \qquad (2)$$

it is possible to attribute the occurrence of extremes in the present-day climate to dynamic and thermodynamic ($\Delta\text{MFC}_{td} = \Delta\text{MFC}_{tot} - \Delta\text{MFC}_{dyn}$) contributions. In equation (2), subscript CTL corresponds to all days of the CTL simulation and EX only to the 24 h periods (9:00 to 9:00 UTC) associated with night-time precipitation above the 99th percentile (00:00 to 9:00 UTC). As such, the averaged specific humidities during extremes $\overline{q_{EX}}$ and the entire time period $\overline{q_{CTL}}$ were used to rescale the moisture content during extremes to climatological values. Equation (2) thus computes $\Delta\text{MFC}$ assuming no changes in atmospheric moisture content, that is, taking only circulation changes into account (dynamic contribution). In addition, we used this framework to identify the drivers of future changes in extreme precipitation: in this case we only considered the 24 h periods associated with extreme night-time precipitation in the HIS and FUT simulations, respectively.

**CORDEX ensemble analysis.** The public domain RCP8.5 ensemble established by the Coordinated Regional climate Downscaling Experiment (CORDEX) for Africa currently consists of 16 members, from which we selected the nine members that compute the two-way lake–atmosphere exchange interactively with a lake model (in casu FLake; Supplementary Table 1). Daily precipitation sums of these nine members are available from 1951 to 2100 at 0.44° (∼50 km) resolution and served as a basis to calculate return period and binned precipitation changes after nearest-neighbour remapping to the COSMO-CLM² grid.

**Data availability.** All materials that have contributed to the reported results are available upon request, including code and the COSMO-CLM² model output

(26 TB). CORDEX-Africa simulations are available at https://esgf-data.dkrz.de/, ERA-Interim data at http://www.ecmwf.int/en/research/climate-reanalysis/era-interim and TRMM observations at http://pmm.nasa.gov/data-access/downloads/trmm.

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

## Acknowledgements

We acknowledge the CLM community (http://www.clm-community.eu) for developing COSMO-CLM² and making the model code available, and Hans-Jürgen Panitz for providing the lateral boundary conditions. In addition, we are grateful to the World Climate Research Programme (WRCP) for initiating and coordinating the CORDEX-Africa initiative, to the modelling centres for making their downscaling results publicly available through ESGF, to ECMWF for providing access to ERA-Interim, and to NASA and JAXA for developing the TRMM-3B42 data set. We particularly thank Fabien Chatterjee, Matthias Demuzere, David Docquier, Niels Souverijns and Kristof Van Tricht for their useful suggestions. W.T. was supported by a PhD fellowship from the Research Foundation Flanders (FWO) and an ETH Zurich postdoctoral fellowship (Fel-45 15–1). S.L. was supported by an FWO postdoctoral fellowship. The Belgian Science Policy Office (BELSPO) is acknowledged for the support through the research project EAGLES (CD/AR/02A). Computational resources and services used for the COSMO-CLM² simulation were provided by the VSC (Flemish Supercomputer Center), funded by the Hercules Foundation and the Flemish Government—department EWI.

## Author contributions

W.T., N.P.M.v.L., E.L.D. and S.I.S. designed the study. W.T. conducted the COSMO-CLM² simulations, performed all model analyses and wrote the manuscript. K.B. developed the OT detection algorithm and applied it to East Africa. W.T., S.L. and K.B. analysed these data. All authors commented on the manuscript.

## Additional information

**Competing financial interests:** The authors declare no competing financial interests.

