## [Peer Review File · Nature Communications]

Reviewers' comments:

Reviewer #1 (Remarks to the Author):

Summary of results:

This paper examines the possible increase in severe weather near Lake Victoria, Africa, as a result of climate change. The authors fuse model and satellite data to determine that extreme precipitation events will become more likely in the case of the high emission future climate projection.

Originality:

Overall I think the paper presents an original idea, and I particularly like the joint use of model and satellite data. To my knowledge no similar studies currently exist. However, the paper does overstate its originality regarding the satellite data used within. On line 43 of page 2 the authors state that their work has established the first severe thunderstorm climatology for E.Africa. This is not the case, Proud (2015, doi: 10.1002/qj.2410) shows something similar. Indeed, it specifically mentions the difference in day/night overshooting tops over Lake Victoria. Due to the crossover in the work I suggest that this paper is acknowledged in some way. In addition, whilst not observing overshooting tops there has been prior thunderstorm climatologies produced from TRMM data. One such paper has actually been referenced in this manuscript (ref: [14]).

Data:

Both the data used and the methodology presented seem well thought-through and appropriate. Care has been taken to discuss limitations of the climate model and its accuracy is discussed within the paper. The comparison to the low-res climate model is especially useful in this regard. I am interested to know, though, why the authors chose the high emission scenario alone, and did not look at one of the lower (possibly more likely) scenarios. It would be good for this to be discussed and also specifically mentioned in the abstract (it's currently buried in the middle of the article).

Uncertainties:

The data/model uncertainties are discussed in the supplementary material. I would like to see an expanded discussion of them in the main text, though. Whilst it is obvious that uncertainties have been considered during this work there is sparse mention of them in the main text. Quantitative data are presented without the corresponding uncertainties, for example. See also my comments below regarding manuscript clarity.

Conclusions:

I think that the conclusions drawn by this manuscript are appropriate and robust. I see no problems with them.

References: Aside from the aforementioned references to prior OT/thunderstorm studies in E. Africa I think that the references are appropriate and that credit is given when needed.

Clarity of text: The article is well-written. My only concern in this respect is that throughout the manuscript the model forecasts are presented as fact. For example, page 1 line 15 states: "Here we show that Lake Victoria is a hotspot..."

"is" should be suffixed with "predicted to be". Other examples are seen, for example, on page 2 line 65 and page 3 line 74. Whilst I appreciate that the results from the models appear robust one should remain careful not to present them as an absolute certainty.

A minor point is that on the second line of the abstract the word "presumably" is used. This seems inappropriate for a scientific manuscript. Please edit this sentence.

Reviewer #2 (Remarks to the Author):

I begin my answering the general questions, as requested by the editor. Detailed major and minor concerns follow.

What are the major claims of the paper?

This paper uses 'high-resolution' climate model simulations over the East Africa to demonstrate that whilst regional mean precipitation is expected to decrease under climate change, the severity of precipitation extremes are expected to increase over Lake Victoria. The results demonstrate that the mechanism for these precipitation changes is the modification of moisture convergence over the lake, through differing changes to the dynamical (wind/advection) and thermodynamical (humidity) contributions. A surprising result is that increases in evaporation of moisture from the lake's surface in a future warmer temperatures makes only a very small contribution to the precipitation changes.

The main motivation for the paper is the safety of fisherman on the lake. Many die each year on the lake as a consequence of the extreme storms. However, as I discuss below the main hazard for fisherman on the lake is the wind-driven waves, not the precipitation, which devalues the paper's motivation.

The paper addresses what is a very regional issue – there is no attempt to discuss the wider implications (in a geographical sense), although it is my opinion that it is not possible to do so with the model products used in the study. The paper says that the results are motivation to develop an early warning system for severe storms on the lake. However, I know that such a system has already been developed and is operational in the region (see major concern below).

Are they novel and will they be of interest to others in the field?

To my knowledge the major results outlined above have not been demonstrated in previous peer-reviewed literature.

Is the work convincing, and if not, what further evidence would be required to strengthen the conclusions?

Overall, the work is impressive, interesting and convincing but there are a number of issues with the way the results are presented, there is a lack of detailed explanation for some aspects of the work and some further analysis is necessary. See major and minor concerns below.

On a more subjective note, do you feel that the paper will influence thinking in the field?

Yes, the paper will influence thinking in the fields of high-resolution climate modelling and East African meteorology. The paper demonstrates the value of high-resolution simulations, albeit that still employ a convection parameterisation. It will be a strong motivator for scientists to run cloud-resolving (i.e. without a convection parameterisation) future climate simulations over the region. The work demonstrates to climate scientists that it is crucial to adequately represent mesoscale circulations in models to understand how rainfall will change in the future.

Recommendation

Overall I think the paper contains some very valuable research on extreme precipitation and storms over the Lake but the paper is not publishable in its current form. It suffers including too much information and may be improved by splitting into two separate papers or by removing some of the model simulations from the analysis.

Major concerns

The main motivation for the paper is that many fisherman die on the lake each year due to the severe storms. It is, however, the wind-driven waves from the strong storm updrafts that kill the fisherman, not the heavy precipitation^{1,2}. The paper does not analyse changes in wind at all. Either the paper should analyse wind extremes or this motivation should be modified or removed. Is there a simple correlation between rainfall intensity and storm downdrafts/wind in your model simulations? The results of the paper are, however, still valuable; water security is a major problem in East Africa (although I am not familiar with how important local rainfall rates over the lake are for the total lake's volume – I assume the lake is used as a major source of water to the local populations for drinking, irrigation etc).

The World Meteorological Organisation has already lead an effort to develop a severe storms early warning system for the region using a high-resolution (cloud-resolving) operational forecast system run by the UK Met Office. The Met Office give the local Met Agencies access to the data, who produce forecasts and issue severe storm warns to fisherman via text message^{1,2}. This system has been operational for a number of years.

I found that reading the full paper (main article plus supplementary material) hard going. This is partly because the material is spread over three different sections (main, supplementary, methods) and partly due to the sheer volume of material, model simulations and analysis presented. There are 15 figures in total which is a large number even for a standard length paper. I don't know if there is a word limit for the supplementary material but, even given the current length of the paper, I found the explanation of many of the supplementary figures insufficient e.g. Figures S6, S7 and S8 get a single sentence each. My feeling is that there is too much material here for a single Nature-style paper. I suggest that the results are split into two: perhaps one standard journal paper covering the model evaluation of the present-day simulations and then a shorter Nature-style paper covering the exciting future climate results. I know this has worked well for previous evaluations of high-resolution future climate model simulations and I don't think the initial paper would take away anything from the novelty of the second. The other option might be to strip out the CORDEX and ERAI part of the analysis and focus solely on the COSMO-CLM simulations. I think a more coherent, shorter paper could also be achieved that way.

The schematic presented in Figure 4 needs improvement. In my view the purpose of this kind of schematic is to convey the main mechanisms for rainfall changes quickly and simply to the reader. In its current form it does not achieve this. The variation in blue shading on the circulation arrows is not clear enough. The difference between the 'extreme' and 'mean' rainfall in the cartoon is not clear. What is represented by the blue and red arrows is not obvious – I needed to read the full paragraph caption to understand what these arrows meant. It might be easier to just use some words in each panel to talk about relative increases/decreased in thermodynamical and dynamical contributions.

I suggest adding a table summarising the simulations used to the supplementary material, including basic details of the CORDEX simulations. I found the explanation of these confusing until I had drawn up my own table. The NOL (No Lake) experiment is only mentioned very briefly so is hidden in the details.

I suggest being upfront in the main body of text about what the horizontal resolution of COSMO-CLM is and that the COSMO-CLM simulations have parameterised convection (i.e. are not cloud-resolving, which could be expected as 'high-resolution' these days). This detail is hidden somewhere in the

supplementary material. Using just 'high-resolution' is not adequate because this means different things to different people.

Can the authors be more clear (in the response to reviewers) about what the added value of the COSMO-CLM simulations really is? It seems as though they produce the same results as the CORDEX models (Figures S7,9,10). The exception is the rainfall pdf's in Figure S3, but this doesn't seem to impact the models' ability to represent the mechanisms that control the precipitation. Both CORDEX and COSMO-CLM both employ a convection parameterisation, which likely produces a whole host of well-known biases in rainfall (too much light rain, incorrect diurnal cycle, lack of storm propagation). Does the increase in resolution from CORDEX (25km?) to COSMO-CLM 7km really make that much difference? Related to this is the model bias in Figure S4g. Why does COSMO-CLM not get the band of extreme precipitation over the land around the lake that is seen in TRMM? The paper states that it is due to the incorrect diurnal cycle through the convection parameterisation but it is not as if the extremes are appearing at the wrong time of day – they are missing completely.

In general there is a lack of discussion about the relative importance of mesoscale vs synoptic scale controls on the moisture convergence and precipitation. This can be illustrated as two separate issues:

1. What impact do possible biases from the lateral boundary conditions from the GCM have within the CORDEX-CLM domain for both the HIS and FUT simulations? Why was the HIS COSMO-CLM simulation not evaluated alongside the CTL simulation? This would determine if the lateral boundary conditions from the GCM are a large source of bias within the domain.
2. I am not convinced that what you define as a mesoscale control is definitely that. Couldn't your mesoscale control on moisture actually be synoptic??

I failed to understand exactly how the plots S7, S8 and 2c,d were created and what they show. This is partly because there is a lack of explanation for them in the text. I am unfamiliar with this particular type of plot and it wasn't clear to me whether P in this case means 'precipitation' or 'percentile'. Can you revise the explanation by expanding it in the supplementary material?

Minor concerns

Main document

L13 and L27. The use of the word 'presumably' is weak. Using 'is estimated at' or similar would be better.

L17 Say upfront here or in the paragraph at L39-55 what is meant by 'high-resolution' in this context.

L43 'first severe thunderstorm climatology for East Africa' I suppose this is true but the climatology looks very much like the mean OLR climatology in other papers³?

L65-66 The statement about storms building up faster is not backed up by evidence in the paper – storm timings are not discussed in the context of the future simulations.

L93-94 Figure 3c doesn't show that moisture convergence increases, only that specific humidity increases. This doesn't necessarily mean moisture convergence (which has a wind component) also increases. Reference some of the other supplementary figures here? Another point related to a major concern above: how do you know from your analysis that it is the land storms that enhance the moisture convergence, rather than some larger-scale synoptic change in the circulation?

L100-101 Where is the evidence for this? Did you plot precipitation extremes vs. lake evaporation?

L106 There is a chicken and egg question here. What causes what? Do the severe storms/precipitation cause high moisture convergence or does the high moisture convergence cause the extreme storms/precipitation?

L109 The interpretation of Figure S11 needs a longer explanation in the supplementary material.

L110 'mesoscale circulation is crucial for triggering extremes' how do you know this is a mesoscale rather than synoptic-scale circulation influence? (related to major concern above)

L125 "nighttime near-surface air temperature will increase" this is not shown in Figure S5, only the daily mean temperatures are shown.

L130 reference of 'supplementary material' – which bit are you referring to?

L200 state the 'the contour' is the red circle in Fig S1 – easier to understand

L212-213 It is not clear to me from this section why you need to scale the moisture content to climatological values. Please justify this briefly here.

L223 Please state what 0.44deg is approximately in km. You could add this detail to the model simulation table.

Caption Figure 2: 'the clear dipole pattern of change indicates an earlier release of extreme precipitation' how does Figure 2b show an earlier release of precipitation? By how many hours is this release earlier?

Supplementary material

L28 I suggest redefining the difference between COSMO-CLM and COSMO-CLM² here as I failed to notice the ², which confused me. This difference is defined in the Methods section – this highlights the difficulty of having the information spread over three sources. Adding a table of model experiments would help.

L68 'results are in close agreement' I disagree with this statement. These plots are on a log scale, which minimises the differences to the reader. I think it would be better to say that the COSMO-CLM² results are closer to the observations than CMIP5 or ERAI precipitation is.

L79 'propagation in westerly direction along the synoptic flow' do you mean the storms are advected at the same speed as the synoptic flow or that the storms propagate faster than the synoptic flow due to the cold pool/regeneration of the convective lifecycle?

L118-129 This text should probably come under a separate heading, as it doesn't really address 'assessing uncertainty'

L136 What are the CORDEX models driven by? I don't *think* this information is anywhere in the paper, though it may be hiding somewhere. If all the CORDEX models are driven by the same GCM it is not a surprise that they behave in a similar way (e.g. Fig S8).

L1146-175 Here you are assessing 99.9th percentile changes, which is one event in three years. The COSMO-CLM simulations are only 29 years long, so this translates to about 9 or 10 events. Is this enough to get statistical significance?

L155 say upfront which models you are doing this analysis for.

L169 why aren't the COSMO-CLM results for the C-C scaling plotted on Figure S10?

L176 By 'moderate extreme' do you mean the 99th percentile? By 'distinct scaling pattern' do you mean the land-lake difference?

Figure S4 if the ERA-Interim data has been remapped onto the COSMO-CLM grid why are the grid sizes difference in Figure S4e-h compared with Figure S4i-l?

References

¹http://www.metoffice.gov.uk/media/pdf/m/r/14_0500_Saving_lives_on_Lake_Victoria.pdf

²<http://www.metoffice.gov.uk/barometer/features/2011-12/saving-lives-on-lake-victoria>

³Chamberlain, J. M., C. L. Bain, D. F. A. Boyd, K. McCourt, T. Butcher, S. Palmer (2014) Forecasting storms over Lake Victoria using a high resolution model, Meteorol. Appl., 21, 419-430.

Reviewer #3 (Remarks to the Author):

Summary

This study has investigated the changing patterns of intense/ extreme precipitation over Lake Victoria Basin (LVB) in equatorial East Africa. LVB is one of the regions which records the highest number thunderstorm days. The study reveals that there is significant intensification of extreme precipitation, consistent in the projections by CORDEX and COSMO-CLM models while there is significant decrease in the mean precipitation over LVB. The authors have used very robust methods of analysis while taking advantage of both model and satellite data. The discussions of the results are quite comprehensive and conclusions consistent with the analysis and model results. I have made the following specific comments and suggestions for improvement.

The driving mechanisms for the intensification of extreme precipitation over the Lake is attributed to increasing thunderstorm activities which are preceded by intense rainfall and storms over land. However, I find this to be somehow inconsistent with previous studies that reveal that over land areas in the entire equatorial eastern Africa region, rainfall has been consistently decreasing over the past decades unlike what the models (including majority of IPCC AR5) projections actually indicate. The trend currently referred to in some studies as the East Africa climate paradox (e.g. Rowell et al., 2015). Therefore the two theories proposed as the primary mechanisms through which increasing storms over land intensify the thunderstorms over the lake (e.g. page. 3, lines 93-97) are not fully supported by the observed decrease in rainfall over land. Based on large scale dynamics of the East Africa (LVB) rainfall (e.g. Nicholson et al. 2000) the decreasing rainfall trends over the region means that there is suppressed large scale horizontal moisture transport and convergence over land areas (may be based on decadal or longer variability).

The second inconsistency which is related to the above is the fact that the results of this study also shows that overlake evaporation (precipitation recycling!) is also not likely to contribute the intensification of thunderstorms over the lake. I also find this as inconsistent given that many studies have consistently shown that the water balance over the lake is dominated by the near balance between ET and Precipitation. Could the models be misrepresenting the contribution of evaporation? If not, despite the sporadic intense storms over land areas that the study show to be the precursor of the intense over-lake thunderstorms, how would the intense thunderstorms be sustained as the land areas are getting drier? This is somehow alluded to on page 4 (lines 113-116), but for projections

which also indicate reduction in moisture convergence by ~3%. What are the large-scale moisture transport mechanisms?

On page 4 (lines 133-145) While the model projects a general decrease in mean precipitation, again it is not clear what are the primary sources of moisture that triggers and sustains extreme thunderstorms over the lake.

On page 4 (line 143) the approximate number of people directly supported by Lake Victoria is 35million (not 3.5 million).

Reviewer #4 (Remarks to the Author):

The authors investigate extreme precipitation in the current climate and its response to climate change over Lake Victoria. They find that extremes will intensify in this region, mainly due to thermodynamic effects (increased atmospheric humidity).

The topic of the paper and the results are interesting, I think that the work shown in this manuscript can be published. But I think that significant clarifications and further investigations are needed to complete the study. I therefore can not recommend publication in the current form.

First my major overall comment is that correlation does not imply causality. Throughout the paper, conclusions are drawn about the causality between 2 variables based on correlation between the 2. I didn't find the arguments always convincing.

For example, the authors find a correlation between the afternoon land breeze and extreme lake precipitation (figure 3d). It is not clear to me how afternoon land breeze can impact nighttime precipitation extremes, what is the physical mechanism? Probably the key variable controlling the dynamic contribution is the temperature difference between land and lake. Of course this is related to the afternoon land breeze as well, but the real physical process and causality is between the land/lake temperature contrast and extreme lake precipitation. Another remark about figure 3d is that the extreme precipitation actually appears to be largely insensitive to the lake breeze (flat curve) except the last decade of precipitation. What precipitation percentile does that correspond to?

Another example is the increased moisture interpreted as being the result of enhanced land storms. I am not a specialist of this region, but I think one could argue the opposite causality: increased land storms being the result of increased moisture, which could be due to another source (large scale advection...). I don't think that the analysis of this manuscript is sufficient to deduce that the afternoon land storms are responsible for, and not caused by, the enhanced moisture. There may be other moisture sources, and both land and lake OT high counts would result from enhanced moisture.

Also I found the presentation and wording not always clear. For example:

- Figure 2 the color bar convention switches between panels a and b.
- Regarding the OT counts (line 46, Methods) from the methods I understand that adjacent OT pixels are identified as being part of the same OT. But line 46 mentions 1,400,000 OTs over the lake alone, for the 9 years investigated that yields more than 400 OTs per 24h. I guess this means that several OTs are found in one thunderstorm, I think that the definition and physical meaning of OT could be clarified (at least in Methods).
- line 62-66: "the spatial pattern indicates an eastward shift... in the future storms develop faster leading to more extreme precipitation in the east" This is speculative, I do not think that the distribution of figure 2 allows you to reach this conclusion. Have the authors quantified the time duration of those thunderstorms, to check whether it is shorter with warming? Alternatively, the speed of the trade winds may change (leading to different speed of advection over the lake). Or the storms could be initiated further east compared to current climate.
- line 109: "enhanced moisture supplied by afternoon land precipitation" How do you know that the moisture is supplied by land precipitation? There could be other moisture sources (in particular large-scale advection).
- Figure 2: are all the panels a-d from the COSMO-CLM simulations? "The clear dipole pattern of change indicates an earlier release of extreme precipitation": I did not find this wording very clear.
- line 108 and fig3c-d: it looks like the dynamic control dominates only the last decade, before that the thermodynamic effect is larger. So which effect dominates may depend on the precipitation percentile considered.

Author's response to reviewers

W. Thiery, E.L. Davin, S.I. Seneviratne, K. Bedka, S. Lhermitte, N.P.M. van Lipzig

The authors would like to thank all reviewers for their dedicated time reviewing the manuscript and for their useful and constructive suggestions. All comments by the reviewers were carefully addressed and the manuscript has substantially benefited from the proposed changes. Here below, we would like to clarify our changes regarding all comments, which are repeated as **bold blue text**.

This response letter contains numbered illustrations and references to these illustrations. Except when indicated explicitly, reference is given to line numbers and figures in the originally submitted manuscript, and not to the new, re-submitted manuscript. To prevent confusion, the figures embedded within this response letter are called *illustrations*. Finally, the following convention is applied to denote modification in the original manuscript: ~~deleted words~~; **added words**.

Reviewer #1

“This paper examines the possible increase in severe weather near Lake Victoria, Africa, as a result of climate change. The authors fuse model and satellite data to determine that extreme precipitation events will become more likely in the case of the high emission future climate projection

I think that the conclusions drawn by this manuscript are appropriate and robust. I see no problems with them.

Aside from the aforementioned references to prior OT/thunderstorm studies in E. Africa I think that the references are appropriate and that credit is given when needed.

Overall I think the paper presents an original idea, and I particularly like the joint use of model and satellite data. To my knowledge no similar studies currently exist.”

We thank reviewer #1 for his/her overall support of our study. Below we address the issues that were raised for improvement of the manuscript.

Comment 1: “However, the paper does overstate its originality regarding the

satellite data used within. On line 43 of page 2 the authors state that their work has established the first severe thunderstorm climatology for E.Africa. This is not the case, Proud (2015, doi: 10.1002/qj.2410) shows something similar. Indeed, it specifically mentions the difference in day/night overshooting tops over Lake Victoria. Due to the crossover in the work I suggest that this paper is acknowledged in some way. In addition, whilst not observing overshooting tops there has been prior thunderstorm climatologies produced from TRMM data. One such paper has actually been referenced in this manuscript (ref: [14]).”

We were not aware of this very recent publication presenting a 5-year OT dataset over the MSG domain and thank the reviewer for bringing it to our attention. To improve the contextualisation of our OT analysis, we included references to Proud (2015), Zipser et al. (2006) and Fensholt et al. (2011) where appropriate and updated the text as follows:

- P2L43: “By applying an OT detection algorithm to Meteosat Second Generation observations (Methods), we establish ~~the first~~ **one of the first** severe thunderstorm ~~climatology~~ **climatologies** for East Africa.”

Comment 2: “Both the data used and the methodology presented seem well thought-through and appropriate. Care has been taken to discuss limitations of the climate model and its accuracy is discussed within the paper. The comparison to the low-res climate model is especially useful in this regard. I am interested to know, though, why the authors chose the high emission scenario alone, and did not look at one of the lower (possibly more likely) scenarios. It would be good for this to be discussed and also specifically mentioned in the abstract (it's currently buried in the middle of the article).”

High-resolution regional climate simulations are computationally extremely expensive due to large number of horizontal grid points and vertical layers as well as the small time step (60 sec in our case). As an illustration, the GCM downscaling simulations presented in this manuscript ran for more than four months (walltime), using ~10% of the total High-Performance Computing infrastructure available at KU Leuven during that time. For this reason, only one climate realisation with one driving GCM and for one emission scenario could be possibly done at this resolution. Note that this is a recurrent limitation of very recent studies investigating climate change at high resolution (e.g. Kendon et al., 2014, Ban et al., 2015, Chan et al., 2015).

Since the IPCC does not attach a likelihood estimate to the different RCP scenarios, there is no objective reason to think that one emission scenario is more likely than the others. The high emission scenario was selected as this scenario is expected to yield the strongest responses in the climate system. This facilitates both the interpretation of the results and helps to unravel the physical mechanisms driving the projected changes. It is also a useful choice as it provides the likely upper bound of the changes which may be expected by the end of the century.

Finally, although not with COSMO-CLM², we actually can investigate the influence of the emission scenario by comparing RCP8.5 and RCP4.5 projections within CORDEX. We find that the choice of the RCP does not influence Lake Victoria's amplifying role on extreme precipitation changes: the amplification of extreme precipitation is 1.6 +/- 0.7 times stronger over Lake Victoria compared to surrounding land under RCP4.5 (compared to 1.8 +/- 1.0 times stronger in RCP8.5), and even 4.6 +/- 1.0 times stronger in case we correct for the average drying (compared to 4.2 +/- 1.6 times stronger under RCP8.5; see Supplementary Fig. S7 for a description of this correction). We included a summary of this analysis in the new section 'assessing uncertainty' in the main text (see response to Comment 3 by reviewer #1). In addition, we updated the abstract as follows (Revised abstract counts 155 words, 150 words is the official upper limit):

- P1L20: "The future increase in extremes over Lake Victoria is about twice as large relative to surrounding land **under a high-emission scenario**, as only over-lake moisture advection is high enough to sustain Clausius-Clapeyron scaling."
- P3L80: "Based on a single high-resolution projection (~7 km) we cannot assess modelling uncertainties **or compare emission scenarios**."
- P6L184: "GCM downscalings were performed for the historical reference period 1978-2010 (HIS) and the future projection 2068-2100 under the high emission scenario RCP8.5 (FUT). **This scenario was chosen as it is expected to facilitate interpretation by yielding a strong climate response and as it provides the likely upper bound of the changes which may be expected by the end of the century.**"

Comment 3: "The data/model uncertainties are discussed in the supplementary material. I would like to see an expanded discussion of them in the main text, though. Whilst it is obvious that uncertainties have been considered during this work there is sparse mention of them in the main text. Quantitative data are presented without the corresponding uncertainties, for example. See also my comments below regarding manuscript clarity."

In the revised manuscript, the paragraph "Assessing uncertainty" in the Supplementary Information is split into two parts. We renamed the first part "Correlation versus causality"

and expanded it in response to comment 1 by referee #4. The second part was expanded, moved to the main text and merged with paragraph 5 to yield a more elaborate discussion of the resulting uncertainties (see also comments by reviewer #2). The new subsection reads as follows:

- **P3L80-86: “Assessing uncertainty. Based on a single high-resolution projection (~7 km) we cannot assess modelling uncertainties or compare emission scenarios. With this type of simulations being computationally very expensive, this is a recurrent limitation of studies investigating climate change at high resolution (Kendon et al., 2014; Chan et al., 2015; Ban et al., 2015; Prein et al., 2015). By providing ensemble projections at coarser resolution (~50 km), the CORDEX initiative enables uncertainty assessments within the constraints of the quality of both the downscaling tool and the lateral boundary conditions (Xie et al., 2015). Although some differences occur between the high- and coarse-resolution projections, it is clear that the lake effect on the future precipitation distribution is robust (Fig. 2c-d, Supplementary Fig. S6). This is further confirmed by the fact that the projected response in the coarse-resolution ensemble (Fig. 2d) is to a large extent independent of the driving global model. In particular, every CORDEX simulation projects a reduction in over-lake precipitation for all bins below the 90th percentile and an amplification of the increase in the highest bins, thereby corroborating the high-resolution model (Fig. 2). Comparison of the coarse-resolution RCP8.5 and RCP4.5 ensembles moreover demonstrates that the choice of the emission scenario does not influence Lake Victoria's amplifying role on extreme precipitation changes.**

At the same time COSMO-CLM² clearly outperforms all CORDEX models as well as a state-of-the-art reanalysis in terms of precipitation representation, underlining the benefits of enhanced resolution and use of a lake model for climate simulations over the region (Supplementary Information) (Akkermans et al., 2014; Thiery et al., 2014a, 2014b, 2015; Docquier et al., 2016). Decreasing the horizontal grid spacing to convection-permitting scales (below ~4 km) would most likely improve the skill of our climate simulations even more, since the convection parameterisation employed in the high-resolution model still entails a number of limitations (Lauwaet et al., 2010; Kendon et al., 2012; Ban et al., 2014, 2015; Prein et al., 2015; Brisson et al., 2016). Overall these findings highlight the need for running coordinated high-resolution projections to quantify local climate change in regions with a particular dynamical regime (Kendon et al., 2014).”

In addition we now included uncertainty ranges in the quantification of the amplification effect. We first subtracted/added one standard deviation from the mean of the highest bin

and from that computed the “min”/“max” amplification effect. The uncertainty range was subsequently obtained as the maximum difference between the mean amplification effect on the one hand and the “min”/“max” amplification effect on the other hand. The ranges are included in the revised manuscript and their computation is now explained in the methods (see also response to comment 9 by reviewer #2).

Comment 4: “Clarity of text: The article is well-written. My only concern in this respect is that throughout the manuscript the model forecasts are presented as fact. For example, page 1 line 15 states: “Here we show that Lake Victoria is a hotspot...” => “is” should be suffixed with “predicted to be”. Other examples are seen, for example, on page 2 line 65 and page 3 line 74. Whilst I appreciate that the results from the models appear robust one should remain careful not to present them as an absolute certainty.”

- **P1L15:** “Here we show that Lake Victoria is **projected to be** a hotspot of future extreme precipitation intensification, using new satellite-based observations, the first high-resolution climate projection for the African Great Lakes and coarser-scale ensemble projections.”
- **P2L65:** “In the future, storms **are projected to** develop faster leading to more extreme precipitation in the east.”
- **P3L74:** “In other words, very intense storms ~~will~~ **are projected to** become more frequent in the future over Lake Victoria.”

Comment 5: “A minor point is that on the second line of the abstract the word “presumably” is used. This seems inappropriate for a scientific manuscript. Please edit this sentence.”

The original wording was chosen since the exact number of fatalities is subject to high uncertainty. To quote the Red Cross World disaster report: “No proper figures are available for the number of incidents in which people drown because wind and large waves capsize or destroy their boats. But all the relevant agencies consider that it is appropriate to assume that between 3,000 and 5,000 people die each year” (Red Cross, 2014). We adapted the abstract by removing the word ‘presumably’ from the abstract and presenting the number of fatalities in the appropriate words in the first paragraph:

- **P1L12:** “Weather extremes have harmful impacts on communities around Lake Victoria, where ~~presumably~~ thousands of fishermen die every year because of intense nighttime thunderstorms.”

- P1L27: “According to the International Red Cross², presumably **The International Red Cross assumes that** 3000 to 5000 fishermen die every year on the lake², ~~thereby~~ **by which it** substantially ~~contributing~~ **contributes** to the global death toll from natural disasters.”

Reviewer #2

“This paper uses ‘high-resolution’ climate model simulations over the East Africa to demonstrate that whilst regional mean precipitation is expected to decrease under climate change, the severity of precipitation extremes are expected to increase over Lake Victoria. The results demonstrate that the mechanism for these precipitation changes is the modification of moisture convergence over the lake, through differing changes to the dynamical (wind/advection) and thermodynamical (humidity) contributions. A surprising result is that increases in evaporation of moisture from the lake’s surface in a future warmer temperatures makes only a very small contribution to the precipitation changes.

The main motivation for the paper is the safety of fisherman on the lake. Many die each year on the lake as a consequence of the extreme storms. However, as I discuss below the main hazard for fisherman on the lake is the wind-driven waves, not the precipitation, which devalues the paper’s motivation.

The paper addresses what is a very regional issue – there is no attempt to discuss the wider implications (in a geographical sense), although it is my opinion that it is not possible to do so with the model products used in the study. The paper says that the results are motivation to develop an early warning system for severe storms on the lake. However, I know that such a system has already been developed and is operational in the region (see major concern below).

To my knowledge the major results outlined above have not been demonstrated in previous peer reviewed literature.

Overall, the work is impressive, interesting and convincing but there are a number of issues with the way the results are presented, there is a lack of detailed explanation for some aspects of the work and some further analysis is necessary. See major and minor concerns below.

Yes, the paper will influence thinking in the fields of high-resolution climate modelling and East African meteorology. The paper demonstrates the value of high-resolution simulations, albeit that still employ a convection parameterisation. It will be a strong motivator for scientists to run cloud-resolving (i.e. without a convection parameterisation) future climate simulations over the region. The work demonstrates to climate scientists that it is crucial to adequately represent mesoscale circulations in models to understand how rainfall will change in the future.

Overall I think the paper contains some very valuable research on extreme precipitation and storms over the Lake but the paper is not publishable in its current form. It suffers including too much information and may be improved by splitting into two separate papers or by removing some of the model simulations from the analysis.”

We thank the reviewer for raising important concerns. We have thoroughly revised the manuscript to address these issues and to further clarify the methodology. Detailed answers are provided below.

Comment 1: “The main motivation for the paper is that many fisherman die on the lake each year due to the severe storms. It is, however, the wind-driven waves from the strong storm updrafts that kill the fisherman, not the heavy precipitation^{1,2}. The paper does not analyse changes in wind at all. Either the paper should analyse wind extremes or this motivation should be modified or removed. Is there a simple correlation between rainfall intensity and storm downdrafts/wind in your model simulations? The results of the paper are, however, still valuable; water security is a major problem in East Africa (although I am not familiar with how important local rainfall rates over the lake are for the total lake’s volume – I assume the lake is used as a major source of water to the local populations for drinking, irrigation etc).”

In reality extreme precipitation and wind gusts are closely linked to each other. Satellite observations confirm the relation between intense precipitation, strong surface wind gusts and OTs (see Bedka, 2011 for an overview). Moreover, observations from the only operational AWS over the African Great Lakes also demonstrate the link (Illustration 1). At AWS Kivu, we find a statistically significant spearman rank correlation of 0.37 for nighttime maximum wind speed versus nighttime maximum precipitation (nights without precipitation excluded). We note however that Lake Kivu is much smaller compared to Lake Victoria, and that our weather station is located only ~2 km offshore. Hence an even stronger link can be expected over Lake Victoria. Finally, the link was also reported in an early study of Lake Victoria (Carpenter, 1922).

Illustration 1 : left: AWS Kivu. Right: Nighttime maximum precipitation (30 min accumulated) versus nighttime maximum nighttime wind velocity (instantaneous) as observed over AWS Kivu from 09/10/2012 to 10/04/2016. Nights with maximum precipitation values below 1 mm/30min

were excluded. AWS Kivu is the only scientific Automatic Weather Station (AWS) currently operational over one of the African Great Lakes. Wim Thiery installed this station ~2 km offshore the city of Gisenyi (Rwanda) in October 2012 within the framework of the EAGLES project (<http://www.eagles-kivu.be/>).

Although also the model shows a statistically significant positive rank correlation (0.29), we do not believe that Regional Climate Model simulations with parameterized convection are the most suitable tool to investigate strong wind gusts associated with lake thunderstorms. As we mention further on, increasing the model resolution to convection-permitting horizontal scales will allow for more in-depth analyses of cold pool dynamics and severe surface winds (e.g. Lauwaet et al., 2012; Mazon and Pino, 2013, 2014, 2015). Moreover, our simulations contain only 3-hourly averaged wind velocity fields as output, so we cannot investigate wind gusts directly (that is, maximum *instantaneous* winds).

In conclusion, while the link between intense precipitation, wind gusts and lake thunderstorms is clear, we believe that intense precipitation is a better proxy for thunderstorm intensity in RCM runs with parameterised convection. We adapted the introduction section in this direction. We also included a new paragraph 'Assessing uncertainty' where the potential of convection-permitting modelling is discussed in more detail (see also response to comment 3 by reviewer #1):

- **P3L80-86: "Assessing uncertainty. Based on a single high-resolution projection (~7 km) we cannot assess modelling uncertainties. With this type of simulations being computationally very expensive, this is a recurrent limitation of studies investigating climate change at high resolution (Kendon et al., 2014; Chan et al., 2015; Ban et al., 2015; Prein et al., 2015). By providing ensemble projections at coarser resolution (~50 km), the CORDEX initiative enables uncertainty assessments within the constraints of the quality of both the downscaling tool and the lateral boundary conditions (Xie et al., 2015). Although some differences occur between the high- and coarse-resolution projections, it is clear that the lake effect on the future precipitation distribution is robust (Fig. 2c-d, Supplementary Fig. S6). This is further confirmed by the fact that the projected response in the coarse-resolution ensemble (Fig. 2d) is to a large extent independent of the driving global model. In particular, every CORDEX simulation projects a reduction in over-lake precipitation for all bins below the 90th percentile and an amplification of the increase in the highest bins, thereby corroborating the high-resolution model (Fig. 2). Comparison of the coarse-resolution RCP8.5 and RCP4.5 ensembles moreover demonstrates that the choice of the emission scenario does not influence Lake Victoria's amplifying role on extreme precipitation changes.**

At the same time COSMO-CLM2 clearly outperforms all CORDEX models as well as a state-of-the-art reanalysis in terms of precipitation representation, underlining the benefits of enhanced resolution and use of a lake model for climate simulations over the region (Supplementary Information) (Akkermans et al., 2014; Thiery et al., 2014a, 2014b, 2015; Docquier et al., 2016). Decreasing the horizontal grid spacing to convection-permitting scales (below ~4 km) would most likely improve the skill

of our climate simulations even more, since the convection parameterisation employed in the high-resolution model still entails a number of limitations (Lauwaet et al., 2010; Kendon et al., 2012; Ban et al., 2014, 2015; Prein et al., 2015; Brisson et al., 2016). Overall these findings highlight the need for running coordinated high-resolution projections to quantify local climate change in regions with a particular dynamical regime (Kendon et al., 2014).”

Comment 2: “The World Meteorological Organisation has already lead an effort to develop a severe storms early warning system for the region using a high-resolution (cloud-resolving) operational forecast system run by the UK Met Office. The Met Office give the local Met Agencies access to the data, who produce forecasts and issue severe storm warns to fisherman via text message^{1,2}. This system has been operational for a number of years..”

We recognise the long-standing efforts of the UK Met Office to provide meteorological forecasts for the region by running the 4 km ‘Lake Victoria Model’ in NWP mode, and are aware of the recent domain expansion to include a larger region of East Africa and the plans to couple FLake to the UM to improve performance of lake regions (Dr. Carolina Bain and Dr. Gabriel Rooney, pers. comm.).

We believe that an NWP and a satellite-based early warning system may be complementary in providing information to local meteorological services and allowing them to issue reliable warnings. We adapted the manuscript to incorporate this information and included the reference to Chamberlain et al. (2012):

- P1L31: ~~“Despite the long-known bad reputation of Lake Victoria (Carpenter, 1922), operational early warning systems to protect local communities are currently lacking. This partly results from the limited the understanding of the drivers of these extreme thunderstorms remains limited (Thiery et al., 2015).”~~
- P5L153: ~~“At the same time, our findings mark an opportunity for developing an a satellite-based early warning system for hazardous thunderstorms over Lake Victoria. Such a warning system could substantially reduce the vulnerability of local fishing communities and would derive predictions from the strong afternoon controls on nighttime thunderstorms~~ **Warning systems deriving predictions from the strong afternoon controls on nighttime thunderstorms (Fig. 3) have the potential to substantially reduce the vulnerability of local fishing communities. This would complement ongoing efforts, in particular by the UK Met Office (Chamberlain et al., 2014), to provide storm warnings for the region based on numerical weather prediction.”**

Comment 3: “I found that reading the full paper (main article plus supplementary material) hard going. This is partly because the material is spread over three different sections (main, supplementary, methods) and partly due to the sheer volume of material, model simulations and analysis presented. There are 15 figures in total which is a large number even for a standard length paper. I don’t know if there is a word limit for the supplementary material but, even given the current length of the paper, I found the explanation of many of the supplementary figures insufficient e.g. Figures S6, S7 and S8 get a single sentence each. My feeling is that there is too much material here for a single Nature-style paper. I suggest that the results are split into two: perhaps one standard journal paper covering the model evaluation of the present-day simulations and then a shorter Nature-style paper covering the exciting future climate results. I know this has worked well for previous evaluations of high-resolution future climate model simulations and I don’t think the initial paper would take away anything from the novelty of the second. The other option might be to strip out the CORDEX and ERAI part of the analysis and focus solely on the COSMO-CLM simulations. I think a more coherent, shorter paper could also be achieved that way.”

As advised by the Nature Communications Editor, rather than dividing the manuscript in two parts, the full paper (main text plus supplementary information) may be shortened by removing unnecessary figures and material. This also renders the storyline more coherent.

We therefore decided to remove figures S6, S7 and S8 as well as references to these figures. Thereby the total number of figures reduces from 15 to 12 (4 in the main text). Statements made in the text about these figures remain valid without these references. A reference to Descy et al. (2015) was included since a variant of Fig. S6 is also shown there.

In addition, textual coherency is improved by relocating several text blocks and removing text where possible. For instance, some of the material presented in the supplementary information was moved to the main manuscript, in particular the description of the correction for the average drying and the description of the binning procedure (to methods) as well as part of our uncertainty assessment (to new section ‘Assessing uncertainty’ in the main text¹; see our response to Comment 3 by reviewer #1). We also removed the reference to the absence of evaporation driving changes in mean precipitation:

- ~~P4L129: “Again the average precipitation decrease is not due to changes in lake evaporation, since it increases by 9% (Supplementary Information).”~~

¹ These changes are possible since Nature Communications has a word limit of 5000 words for the main text (1774 words in the original submission).

This study is not focusing on model evaluation. In fact, several evaluations of our simulations are already published in recent studies (Thiery et al., 2014a, 2014b, 2015; Docquier et al., 2016; see also comment 7 by reviewer #2). Hence our decision to describe additional model evaluation only in the supplementary material. Readers interested in the model evaluation, in particular regarding extreme precipitation, can find all the information there, while we avoid these analyses to blur the main findings of our paper. However, to shorten the paper we decided to remove the short summary of this analysis from the method section:

- ~~P1L31: “The COSMO-CLM² model system demonstrates excellent performance for the present day climate of this region (Akkermans et al., 2014, Thiery et al., 2015, Docquier et al., 2016) and FLake was judged suitable for application to the African Great Lakes (Thiery et al., 2014a, Thiery et al., 2014b). Additional model evaluation moreover establishes the ability of COSMO-CLM² to well represent the precipitation distribution and the diurnal cycle of extremes over Lake Victoria and its surroundings (Supplementary Information). The improved performance relative to ERA-Interim and the CORDEX simulations is mainly due to the inclusion of a lake model and the high horizontal grid resolution, effectively resolving individual lakes and complex topography. These improvements are unprecedented for climate simulations in this region.”~~

Overall, we have made an effort to respond to this concern of reviewer #2 to improve consistency and readability of the paper (by removing 3 supplementary figures, reordering and merging text blocks and removing text where possible). At the same time we think that it is very difficult to reduce the manuscript any further without losing key elements of the results we wish to convey through this paper.

For instance, due to the large computational cost of high-resolution climate simulations, we have performed only one single climate realisation. In that respect the CORDEX ensemble is a useful addition, as it enables a verification of our findings. In this particular case, the ensemble confirms that the projections of the high-resolution model are robust. We also note that Reviewer #1 particularly appreciates these analyses (see Comment 2). Therefore we decided keep them as part of the paper.

Comment 4: “The schematic presented in Figure 4 needs improvement. In my view the purpose of this kind of schematic is to convey the main mechanisms for rainfall changes quickly and simply to the reader. In its current form it does not achieve this. The variation in blue shading on the circulation arrows is not clear enough. The difference between the ‘extreme’ and ‘mean’ rainfall in the cartoon is not clear. What is represented by the blue and red arrows is

not obvious – I needed to read the full paragraph caption to understand what these arrows meant. It might be easier to just use some words in each panel to talk about relative increases/decreased in thermodynamical and dynamical contributions.”

We enhanced the contrast in the rainfall pictograms (mean vs. extremes). In addition, we replaced the purple and red arrows by clarifying text (and removed the legend). Since the variation in blue shading is not clear, we decided to use one dark blue shading on all panels. This is possible since the change in moisture content is now clear from the text. The result is shown in Illustration 2.

Illustration 2 : Updated Fig. 4.

- Caption Fig. 4: “Processes controlling nighttime precipitation extremes and climate change over Lake Victoria. a, In the present-day climate, local evaporation and net moisture flux convergence (MFC; Methods) along the land breeze both contribute to nighttime precipitation generation over Lake Victoria. ~~The width of the circulation cells represents the flow strength (dynamic contribution) and the blue color intensity the atmospheric moisture content (thermodynamic contribution).~~ b, Climate change

simulations project a decrease in average precipitation, despite enhanced lake evaporation. Future **mesoscale circulation** changes ~~in lake induced mesoscale circulation~~ impeding thunderstorm development are responsible for this decrease. c, Present-day precipitation extremes are **associated with increased MFC, of which 74% is explained by enhanced atmospheric convergence and the remaining 26% predominantly controlled by dynamical conditions favouring strong MFC (74%) and further aided** by enhanced moisture content of advected air masses (26%; Fig. 3). d, The future intensification of precipitation extremes is amplified over Lake Victoria compared to surrounding land and entirely due to higher moisture content of converging air masses. ~~Notched arrows reflect the relative magnitude of dynamic (purple) and thermodynamic (red) controls on the net MFC increase.~~”

Comment 5: “I suggest adding a table summarising the simulations used to the supplementary material, including basic details of the CORDEX simulations. I found the explanation of these confusing until I had drawn up my own table. The NOL (No Lake) experiment is only mentioned very briefly so is hidden in the details.”

We inserted the following table into the Supplementary Information:

Table S1: **Overview of COSMO-CLM² and CORDEX model experiments.** For each experiment the employed Regional Climate model (RCM), lateral boundary conditions (LBC), analysis period (excluding spin-up), horizontal grid resolution and lake parameterisation scheme (if any) is mentioned. COSMO-CLM² simulations start three years prior to the first analysis year. The CORDEX-EVAL simulations were only used for model evaluation purposes (Fig. S3). CORDEX simulations 10-16 (highlighted in red) were omitted from the analysis since they were not coupled to a lake model. The FUT simulation was conducted for RCP8.5, whereas CORDEX simulations 1-16 are available both for RCP4.5 and RCP8.5 for the future period. The latter were used both for transient and time slice analyses (Clausius-Clapeyron scaling and comparison to COSMO-CLM², respectively).

Experiment name	RCM	LBC	Analysis period	Resolution	Lake model?
CTL	COSMO-CLM ²	ERA-Interim	1999-2008	0.0625°	FLake
NOL	COSMO-CLM ²	ERA-Interim	1999-2008	0.0625°	(no lakes)
HIS	COSMO-CLM ²	MPI-ESM-LR	1981-2010	0.0625°	FLake
FUT	COSMO-CLM ²	MPI-ESM-LR	2071-2100	0.0625°	FLake
CORDEX-EVAL 1	COSMO-CLM	ERA-Interim	1999-2008	0.44°	No
CORDEX-EVAL 2	RACMO	ERA-Interim	1999-2008	0.44°	No
CORDEX-EVAL 3	RCA4	ERA-Interim	1999-2008	0.44°	FLake
CORDEX 1	RCA4	CanESM2	1951-2100	0.44°	FLake
CORDEX 2	RCA4	CM5A-MR	1951-2100	0.44°	FLake
CORDEX 3	RCA4	CNRM-CM5	1951-2100	0.44°	FLake
CORDEX 4	RCA4	EC-EARTH	1951-2100	0.44°	FLake
CORDEX 5	RCA4	GFDL-ESM2M	1951-2100	0.44°	FLake
CORDEX 6	RCA4	HadGEM2-ES	1951-2100	0.44°	FLake
CORDEX 7	RCA4	MIROC5	1951-2100	0.44°	FLake
CORDEX 8	RCA4	MPI-ESM-LR	1951-2100	0.44°	FLake
CORDEX 9	RCA4	NorESM1-M	1951-2100	0.44°	FLake
CORDEX 10	COSMO-CLM	CNRM-CM5	1951-2100	0.44°	No
CORDEX 11	COSMO-CLM	EC-EARTH	1951-2100	0.44°	No
CORDEX 12	COSMO-CLM	HadGEM2-ES	1951-2100	0.44°	No
CORDEX 13	COSMO-CLM	MPI-ESM-LR	1951-2100	0.44°	No
CORDEX 14	RACMO	EC-EARTH	1951-2100	0.44°	No
CORDEX 15	HIRHAM5	EC-EARTH	1951-2100	0.44°	No
CORDEX 16	HIRHAM5	NorESM1-M	1951-2100	0.44°	No

Comment 6: “I suggest being upfront in the main body of text about what the horizontal resolution of COSMO-CLM is and that the COSMO-CLM simulations have parameterised convection (i.e. are not cloud-resolving, which could be expected as ‘high-resolution’ these days). This detail is hidden somewhere in the supplementary material. Using just ‘high-resolution’ is not adequate because this means different things to different people.”

We acknowledge the fact that high-resolution means different things to different people. Therefore we have updated the main text and the method section to be more clear about our model set-up:

- **P3L83: “To address this question, we performed the first high-resolution (~7 km grid spacing), coupled lake-land-atmosphere climate projection for the African Great Lakes region with the regional climate model COSMO-CLM², ...”**

- P3L83: “However, COSMO-CLM² clearly outperforms all CORDEX models and a state-of-the-art reanalysis ~~in terms of precipitation representation~~ **thanks to its enhanced resolution (~7 km) and use of a lake model** (Supplementary Information), highlighting...”
- P6L186: “**Moist convection is parameterised by the Tiedtke mass flux scheme (Tiedtke 1989).**”

To our knowledge these simulations are so far still the first 30-year climate projections over Africa at a resolution below 25 km, and they can thus be considered as high-resolution climate simulations, especially in the African climate context.

Comment 7: “Can the authors be more clear (in the response to reviewers) about what the added value of the COSMO-CLM simulations really is? It seems as though they produce the same results as the CORDEX models (Figures S7,9,10). The exception is the rainfall pdf’s in Figure S3, but this doesn’t seem to impact the models’ ability to represent the mechanisms that control the precipitation. Both CORDEX and COSMO-CLM both employ a convection parameterisation, which likely produces a whole host of well-known biases in rainfall (too much light rain, incorrect diurnal cycle, lack of storm propagation). Does the increase in resolution from CORDEX (25km?) to COSMO-CLM 7km really make that much difference? Related to this is the model bias in Figure S4g. Why does COSMO-CLM not get the band of extreme precipitation over the land around the lake that is seen in TRMM? The paper states that it is due to the incorrect diurnal cycle through the convection parameterisation but it is not as if the extremes are appearing at the wrong time of day – they are missing completely.”

Added value of high-resolution climate model simulations is commonly conceived either as (i) the improved ability of the high-resolution simulation to represent the present-day climate, or as (ii) the improved climate sensitivity in the high-resolution simulation. True added value is achieved when both are improving with enhanced resolution.

(i) **Added value in terms of model skill.** The improved ability of COSMO-CLM² to represent the present-day climate of the African Great Lakes region relative to CORDEX and ERA-Interim is clear from four publications which present an extensive evaluation and analysis of FLake and the present-day simulations (CTL and NOL). This added value stems from the enhanced resolution (7 km versus 50 km and ~80 km, respectively), but also because of the improved land surface modelling (using Community Land Model) and the inclusion of a lake model (FLake) in COMSO-CLM²:

- Thiery et al. (2014a, 2014b), showing that the lake model FLake is an adequate tool to represent the lake surface temperatures of the African Great Lakes.
- Thiery et al. (2015), evaluating the CTL (and NOL) simulations for 15 different data products. The results reveal a very strong improvement in CTL compared to ERA-Interim and CORDEX-EVAL1, especially for precipitation and lake surface temperatures.
- Docquier et al. (2016), revealing a very good match between the CTL simulation and QuikSCAT satellite observations of near-surface winds (during the dry season).

Therefore the model evaluation section in the present study was limited to the precipitation distribution and precipitation extremes over Lake Victoria. The evaluation confirms our earlier conclusions that the CTL simulation substantially adds value compared to state-of-the-art climate products (see “Evaluation of COSMO-CLM² for extremes” in the Supplementary information).

It is without doubt that further enhancing the horizontal resolution of the CTL simulation to convection permitting scales would lead to an even more pronounced added value in terms of present-day climate representation. This conclusion clearly emerges from a recent body of literature (e.g. Kendon et al., 2012; Ban et al., 2014; Prein et al., 2015; Brisson et al., 2016).

This potential for further improvement is especially relevant in the context of mesoscale lake-atmosphere interactions. For instance, Lauwaet et al., 2012 showed how the cold pool of Lake Chad interacts with the cold pool of land storms to generate precipitation upwind of the lake. For this study they used short simulations with a convection-permitting mesoscale model. The fact that we don’t see this interaction in our climate runs, and therefore miss the observed precipitation maximum east of Lake Victoria, suggests that the convection parameterisation is responsible for this deficiency in our simulations.

A discussion of the potential of convection-permitting climate simulations is now included in the new section ‘assessing uncertainty’ in the main text of the revised paper (see response to comment 1 by reviewer #2), and we made the following changes in the Supplementary Information:

- P3L84 (SI): “During daytime, COSMO-CLM² underestimates convective precipitation over land, especially east of Lake Victoria where interactions between the synoptic and mesoscale circulation generate enhanced thunderstorm activity (~~Lauwaet et al., 2012~~) (Supplementary Fig. S4g-h). **It is well known that convection parameterisations in regional climate models induce a number of biases in the representation of precipitation, including for instance errors in the diurnal cycle or**

the underestimation of heavy precipitation ~~This is a well known issue in regional climate model simulations with parameterized convection (Kendon et al., 2012; Ban et al., 2014; 2015; Prein et al., 2015; Brisson et al., 2016). The absence of the afternoon precipitation maximum East of Lake Victoria moreover suggests another problem related to convection parameterisation. Using a set of convection-permitting simulations, Lauwaet et al. (2012) showed how the cold pool of convective storms interacts with the cold pool of the Lake Chad to generate precipitation upwind of the lake. Such interactions occurring over Lake Victoria are not captured by the CTL simulation. And We therefore assume that higher resolution simulations with explicitly resolved convection will would probably lead to an improved representation of the diurnal cycle of convection precipitation especially over land surrounding Lake Victoria (Lauwaet et al., 2012; Ban et al., 2014; 2015; Prein et al., 2015).~~"

(ii) **Added value in terms of climate sensitivity.** Here we believe that it is not possible to make statements about the added value of enhanced resolution. The reason for this is that the 50 km parent simulation (CORDEX-EVAL1) is different from the CTL simulation is several aspects besides resolution, notably in terms of land and lake surface representation. Possible differences in climate sensitivity can therefore not be ascribed to resolution only.

As a final note, the resolution of CORDEX is 0.44° or ~50 km. This is now specified both in the main text (P3L80-86) and in the method section (P7L222).

Comment 8: "In general there is a lack of discussion about the relative importance of mesoscale vs synoptic scale controls on the moisture convergence and precipitation. This can be illustrated as two separate issues:

- 1. What impact do possible biases from the lateral boundary conditions from the GCM have within the CORDEX-CLM domain for both the HIS and FUT simulations? Why was the HIS COSMO-CLM simulation not evaluated alongside the CTL simulation? This would determine if the lateral boundary conditions from the GCM are a large source of bias within the domain.**
- 2. I am not convinced that what you define as a mesoscale control is definitely that. Couldn't your mesoscale control on moisture actually be synoptic??"**

1. We expanded the evaluation by including the HIS simulation. Results of this comparison are shown in Illustration 3. It is clear that changing the boundary conditions from ERA-Interim to a free-running Global Climate Model (MPI-ESM-LR) does not affect the added value of COSMO-CLM² relative to ERA-Interim (Fig. S2) and CORDEX (Fig. S3), in fact the

change has on a relatively small influence on COSMO-CLM²'s ability to reproduce the observed precipitation distribution. If any change, HIS outperforms CTL for most frequencies, except for rare events over Lake Victoria. However, as mentioned in the Supplementary Information, the reference dataset (TRMM) also becomes increasingly unreliable towards extremes. Overall, we conclude that the lateral boundary conditions of the GCM are no larger source of bias compared to the reanalysis for precipitation around Lake Victoria. We updated Figure S2 to include the evaluation of HIS, and adapted the manuscript:

- **P3L75:** “Finally, changing the boundary conditions from ERA-Interim (CTL) to a free-running Global Climate Model (HIS) has only a small influence on COSMO-CLM²'s ability to reproduce the observed precipitation distribution (Supplementary Fig. S2), suggesting that this change in boundary conditions is no large source of bias within the domain.”

Illustration 3: Influence of Lateral Boundary Conditions on model skill. Cumulative distributions of (a,b) daily and (c,d) 6-hourly precipitation during 1999-2008 for CTL, HIS and ERA-Interim, expressed relative to the number of wet days (daily precipitation exceeding 1 mm d⁻¹, over Lake

Victoria and all land area within the COSMO-CLM² model domain (Fig. S1), respectively. ERA-Interim output was remapped to the COSMO-CLM² grid using nearest neighbour interpolation.

2. As for the synoptic versus mesoscale controls on moisture convergence, this is very important point. We made an effort to establish whether large-scale or mesoscale processes are responsible for triggering extremes in the present-day climate.

One possible way to test causality in a modelling framework is to isolate a given effect using conceptual experiments. We have done this exercise by conducting a COSMO-CLM² simulation identical to CTL, except that we removed all lakes and replaced them by representative land (NOL, see Supplementary Table 2). If the observed relationships (Fig. 3b-d) in NOL would be identical to CTL, this would mean that the lake has no influence on storm activity in the region, and that the relationship is purely driven by large-scale processes such as moisture transport. If, on the other hand, the relationship would be completely absent in NOL, then the lake would be the sole cause of the obtained correlations.

As explained in the Supplementary Information, the results of this exercise indicate that a relationship between, for instance, afternoon land storm and nighttime lake storms exists in NOL, but that the relationship is much more pronounced in CTL. In particular, the mesoscale lake effect attributes for 74% of the median nighttime lake precipitation on average in each bin and for 43% in the bins above the 90th percentile and 43% in the bins above the 99th percentile (Illustration 4). Therefore we conclude that, while large-scale processes are an important driver of lake storms, the afternoon land storms act as a positive feedback for storm activity.

Illustration 4 : Large-scale versus mesoscale influence on the occurrence of nighttime precipitation over Lake Victoria.

In addition, it is possible that future changes in large-scale circulation and moisture transport will modulate the mesoscale interactions identified in the present study as an important component for understanding local climate change. Given the limited domain of our high-resolution model, this question cannot be addressed. Further research is required to fully understand large scale drivers of local precipitation variability and change in the Lake Victoria basin. This could, for instance, be achieved through surrogate climate scenarios whereby thermodynamic changes play but atmospheric dynamics are not affected (e.g. Schär et al., 1996; van Lipzig et al., 2002).

We therefore made the following changes in the manuscript:

- P1L18: “Land precipitation on the previous day exerts a ~~strong~~ control on nighttime occurrence of extremes on the lake by enhancing atmospheric convergence (74%) and moisture availability (26%).”
- P3L93: “**Large-scale moisture availability contributes to this positive relationship, but alone it cannot explain the observed correlation (Supplementary Information). Land storms therefore act as a positive feedback for the intensity of nighttime lake storms.** These severe land storms **could** impact storm intensity over the lake in two ways. First, they enhance moisture convergence by increasing the near-surface specific humidity (thermodynamic control; Fig. 3c and Supplementary Fig. S11). Second, they modify the lake/land breeze system (Thiery et al., 2015) by cooling the land surface (dynamic control). **In that case** the cold pools of the afternoon storms **act to** reduce gradients in near-surface air temperature between lake and land (**Fig. 3d**), thereby weakening the lake breeze and **possibly also** moisture transport away from the lake (**Fig. 3d**). ~~Moreover, persistence of~~ **If the cold anomaly persists** into the night, **this could** strengthens the land breeze and by that **possibly** stimulates moisture convergence and column instability (**Mazon and Pino, 2013**).”
- P3L109: “**A large fraction (74%)** of this increase can be attributed to dynamical effects, while only 26% is due to the enhanced moisture **content of converging air masses** ~~supplied by afternoon land precipitation~~ (Supplementary Fig. S8 and Table S2).”
- P4L117 (SI): “**Correlation versus causality.** Application of the binning procedure to observations and the CTL simulation highlights a strong relationship between afternoon land precipitation and nighttime lake precipitation over Lake Victoria (Fig. 3). By itself this relationship does not imply causality, in fact, there may be a third factor controlling both land and lake precipitation (e.g. synoptic-scale **moisture transport persistence**). Comparison of the CTL simulation to a simulation whereby all lake pixels have been replaced by representative land (NOL) allows us to separate the influence of Lake Victoria from all other effects, including synoptic-scale **moisture transport persistence** (Thiery et al., 2015). Results of this comparison indicate that the mesoscale lake effect attributes for ~~more than 80%~~ **74%** of the median nighttime lake precipitation in each bin, ~~and for 48%~~ **43%** in the bins above the 90th percentile

and 34% in the bins above the 99th percentile (~~nighttime NOL precipitation was binned according to nighttime lake precipitation from CTL~~). From this analysis we conclude that mesoscale lake effects are the dominant cause of the observed positive relationship, although large-scale processes also contribute to it. From this analysis we conclude that large-scale processes alone cannot explain the occurrence of extremes over Lake Victoria, and that mesoscale lake effects are a positive feedback enhancing nighttime thunderstorm intensity.

Our results also highlighted the importance of changes in local atmospheric moisture content and mesoscale processes for understanding the future intensification of precipitation extremes over Lake Victoria. However, it is possible that large-scale variability and climate change modulate these mesoscale interactions. Given the limited domain of our high-resolution model, this question cannot be fully addressed in the present study. Further research is required to fully understand large scale drivers of local precipitation variability and change in the Lake Victoria basin. This could, for instance, be achieved through surrogate climate scenarios whereby thermodynamic changes play but atmospheric dynamics are not affected (e.g. Schär et al., 1996; van Lipzig et al., 2002).”

Comment 9: “I failed to understand exactly how the plots S7, S8 and 2c,d were created and what they show. This is partly because there is a lack of explanation for them in the text. I am unfamiliar with this particular type of plot and it wasn’t clear to me whether P in this case means ‘precipitation’ or ‘percentile’. Can you revise the explanation by expanding it in the supplementary material?”

We averaged the daily accumulated precipitation over Lake Victoria in the HIS and FUT simulations and binned the two time series (using a 1% bin width and assuming tied ranks). The difference of the present and future bins then yields the blue bars shown in fig. 2c (with positive values indicating that precipitation in a given bin is increasing towards the future). The x-axis label Pbin thus stands for “precipitation bin”. The same procedure was repeated for surrounding land (red bars) and CORDEX (Fig. 2d). A new subsection in the methods now explains this procedure.

Note that we removed Figs. S7 and S8 from the manuscript in response to comment 3 by reviewer #2. In these figures, the same procedure was followed except that the binned change was additionally corrected for the average drying. However, the amplification effect of Lake Victoria on precipitation extremes (which is derived from these plots as the ratio of the highest bin change over lake and land, respectively) is still mentioned in the text (P2L69).

Hence it is still needed to explain the correction of binned change for the average drying. This is now done in the Methods (see also comment 3 by reviewer #1):

- **P6L196: “Data binning and correction for average drying. Binned precipitation changes ΔP_{bin} (Fig. 2c-d) were computed using a 1% bin width and assuming tied ranks. The lake influence on extreme precipitation changes was computed as the ratio between the mean daily precipitation change over lake and land for the highest bin (containing precipitation above the 99th percentile). Uncertainty ranges were derived as the maximum difference between this ratio and the ratio obtained with one standard deviation added or subtracted from the mean, respectively. In addition, the binned change was also corrected for the change in mean precipitation. The correction is performed by subtracting from each precipitation bin change ΔP_{bin} the fractional contribution to the average precipitation change assuming equal weights ($P_{bin} \Delta P/P$). By doing so the integral over all bins becomes zero and only the lake influence on the precipitation distribution is retained.”**

Comment 10: “L13 and L27. The use of the word ‘presumably’ is weak. Using ‘is estimated at’ or similar would be better.”

The original wording was chosen since the exact number of fatalities is subject to high uncertainty. To quote the Red Cross World disaster report: “No proper figures are available for the number of incidents in which people drown because wind and large waves capsize or destroy their boats. But all the relevant agencies consider that it is appropriate to assume that between 3,000 and 5,000 people die each year” (Red Cross, 2014). We adapted the abstract by removing the word ‘presumably’ from the abstract and presenting the number of fatalities in the appropriate words in the first paragraph:

- **P1L12: “Weather extremes have harmful impacts on communities around Lake Victoria, where ~~presumably~~ thousands of fishermen die every year because of intense nighttime thunderstorms.”**
- **P1L27: “According to the International Red Cross², ~~presumably~~ **The International Red Cross assumes that** 3000 to 5000 fishermen die every year on the lake², ~~thereby~~ **by which it** substantially ~~contributing~~ **contributes** to the global death toll from natural disasters.”**

Comment 11: “L17 Say upfront here or in the paragraph at L39-55 what is meant by ‘high-resolution’ in this context.”

Since the revised abstract already counts 155 words (150 words is the official upper limit), we now mention the resolution in the paragraph at L39-55 rather than in the abstract. Moreover, we included a new section in the main text discussing the effect of resolution for our results (see our response to comment 1 by reviewer #2).

Comment 12: “L43 ‘first severe thunderstorm climatology for East Africa’ I suppose this is true but the climatology looks very much like the mean OLR climatology in other papers?”

We agree that the OT pattern presented here is very similar to the OLR climatology presented by Chamberlain et al. (2012). Moreover, also the diurnal cycle of TRMM observed precipitation over Lake Victoria demonstrates a similar pattern (Illustration 5). However, it is important to realise that these products represent different processes: while the OLR and TRMM precipitation represent mean cloud cover and precipitation, the OT dataset (Fig. 1) and TRMM 99th percentile maps (Fig. S4) are a measure for extremes. Comparison of the mean and extreme products also highlight some small differences, for instance that the nighttime extremes are even more confined to the lake area compared to the OLR and precipitation climatologies. For completeness, references to Argent et al. (2014), Chamberlain et al. (2014) and Williams et al. (2014), are now included where appropriate in the manuscript.

Illustration 5: Hourly mean precipitation over the African Great Lakes as observed by the Tropical Rainfall Measuring Mission (TRMM) satellite from 1998 to 2013: (a) 9-15 UTC, (b) 0-9 UTC.

Comment 13: “L65-66 The statement about storms building up faster is not backed up by evidence in the paper – storm timings are not discussed in the context of the future simulations.”

This is a very valid point. Figure 2b shows a spatial shift but this does not imply a temporal change. We therefore investigate a possible change in storm timing by looking at the diurnal

cycle of precipitation in present and future climate conditions, and this both for mean and extremes (Illustration 6).

Illustration 6 : projected future change in the diurnal cycle of (a) mean and (b) extreme precipitation over Lake Victoria.

Results indicate that the diurnal cycle of both mean and extreme precipitation are not changing towards the future. We updated the manuscript to be consistent with these new results:

- ~~P2L65: “The spatial pattern of change indicates an eastward shift of intense precipitation systems (Fig. 2a-b). Today convection initiates in the eastern third part of the lake and intensifies while being advected westwards along the trade winds (Carpenter, 1922; Thiery et al., 2015). In the future, storms are projected to release to develop faster leading to more extreme precipitation more in the eastern part of the lake, leading to an eastward shift of intense precipitation (Fig. 2a-b).”~~

Comment 14: “L93-94 Figure 3c doesn’t show that moisture convergence increases, only that specific humidity increases. This doesn’t necessarily mean moisture convergence (which has a wind component) also increases. Reference some of the other supplementary figures here? Another point related to a major concern above: how do you know from your analysis that it is the land storms that enhance the moisture convergence, rather than some larger-scale synoptic change in the circulation?”

We included the missing reference:

- P3L93-94: “First, they enhance moisture convergence by increasing the near-surface specific humidity (thermodynamic control; Fig. 3c and Supplementary Fig. S11).”**

The second point is addressed in response to comment 8 by reviewer #2.

Comment 15: “L100-101 Where is the evidence for this? Did you plot precipitation extremes vs. lake evaporation?”

An intuitive explanation of the severity of thunderstorms over Lake Victoria being caused by excessive evaporation implies that enhanced lake evaporation would lead to enhanced moisture input into the atmosphere, which would in turn enhance thunderstorm intensity. However, as shown in Illustration 7, this expected positive relationship between afternoon evaporation and nighttime precipitation is not present in the CTL simulation. We thus conclude that lake evaporation does not control the occurrence of extremes.

Illustration 7: Afternoon lake evaporation ET_{lake} binned according to nighttime lake precipitation over Lake Victoria from the COSMO-CLM² control simulation (CTL; 1999-2008). Bin width is 1% and uncertainty bands indicate the interquartile range. Note the logarithmic x-axis.

Comment 16: “L106 There is a chicken and egg question here. What causes what? Do the severe storms/precipitation cause high moisture convergence or does the high moisture convergence cause the extreme storms/precipitation?”

This point is addressed in our response to comment 8 by reviewer #2. Overall, we agree that there is no one-way cause-effect relationship in this case. This was badly worded in the original manuscript. In addition, we believe that it is not possible to establish causality for every possible relationship described in the paper. Therefore we reformulated several statements in more careful wordings.

Comment 17: “L109 The interpretation of Figure S11 needs a longer explanation in the supplementary material.”

We included a new paragraph in the Supplementary information describing Fig. S11 (see also our response to Comment 2 by reviewer #3):

- P5L144 (SI): “Moisture Flux Convergence. By applying the Moisture Flux Convergence framework (MFC) to the COSMO-CLM² output, it is possible to quantify the moisture convergence over Lake Victoria for a given time period (Methods). Supplementary Fig. S8a shows the 24h average moisture fluxes in the CTL simulation. Spatial integration yields an average MFC of $258 \times 10^9 \text{ kg d}^{-1}$ from 1999 to 2008. While the atmospheric column loses moisture towards the west, the moisture gain from the east is larger, resulting in a net convergence of moisture. During 24h periods associated with extreme nighttime precipitation (Supplementary Fig. S8b), MFC more than triples to $807 \times 10^9 \text{ kg d}^{-1}$. The total change (Supplementary Fig. S8c) can subsequently be split up into a dynamic (Supplementary Fig. S8d) and thermodynamic contribution (Supplementary Fig. S8d; see Methods how this is done). Results show that the dynamic component is the dominant contributor (74%) to the increase in MFC during extremes (Supplementary Table S2). Note that the relative contribution to the precipitation change may depend on the considered percentile. Investigating this sensitivity is however beyond the scope of the present study.
The same procedure was applied to quantify the future change in MFC during extremes only (Supplementary Table S2). In this case, the increase in MFC by +27% is entirely due to the thermodynamic component, that is, a higher moisture content of advected air masses instead of mesoscale circulation changes.”

Comment 18: “L110 ‘mesoscale circulation is crucial for triggering extremes’ how do you know this is a mesoscale rather than synoptic-scale circulation influence? (related to major concern above).”

This point is discussed in detail in response to Comment 8 by reviewer #2. By comparing extreme precipitation in the CTL and NOL simulation, we find that lake-induced mesoscale circulation is a necessary component for explaining severe thunderstorm occurrence over Lake Victoria (acting as a positive feedback for synoptic scale drivers of extremes).

Comment 19: “L125 “nighttime near-surface air temperature will increase” this is not shown in Figure S5, only the daily mean temperatures are shown.”

Illustration 8 indicates that surface air temperatures are higher over land compared to water for every time of the day. In fact, the differential warming is even more pronounced at night, hence the weakening of the nighttime land breeze can be expected to be even stronger than the strengthening of the afternoon lake breeze. To not further expand the length of the

manuscript, we did not include this figure in the paper, but removed the erroneous reference to Figure S5.

Illustration 8 : diurnal cycle of projected change in near-surface air temperature over Lake Victoria (Blue curve) and surrounding land (red curve).

Comment 20: “L130 reference of ‘supplementary material’ – which bit are you referring to?”

This is a reference to the subsection ‘Projected climate change for the African Great Lakes region’ in the Supplementary Information, where changes in the mean conditions between HIS and FUT are discussed (including the change in lake evaporation in the last paragraph). References to supplementary information subsections are not further specified in the main text (this appears to be common practice in NPG papers).

Note in this context that a rounding mistake was corrected:

- P4L113 (SI): “a relatively large absolute increase is projected over the lake surfaces where evaporation is already high (+142 mm yr or ~~+10%~~ **+9%**;...”

Comment 21: “L200 state the ‘the contour’ is the red circle in Fig S1 – easier to understand.”

- P7L200: “along the ~~contour~~ **red circle** denoted in”

Comment 22: “L212-213 It is not clear to me from this section why you need to scale the moisture content to climatological values. Please justify this briefly here.”

Without the scaling, the total MFC change rather than the dynamic contribution to the total MFC change would be computed with eq. (2)S. The scaling is needed to bring moisture values during extremes back to the present-day climatology. Computing the MFC change during extremes with these ‘pseudo-climatological’ moisture values (but keeping the circulation of the extreme event) thus yields the contribution of changing wind patterns only (= dynamical contribution).

- **P7L213: “Equation (2) thus computes Δ MFC assuming no changes in atmospheric moisture content, that is, taking only circulation changes into account (dynamic contribution).”**

Comment 23: “L223 Please state what 0.44deg is approximately in km. You could add this detail to the model simulation table.”

Model resolutions are included in the new simulation table (Table S1):

- **P7L223: “at 0.44° (~50 km) resolution ...”**

Comment 24: “Caption Figure 2: ‘the clear dipole pattern of change indicates an earlier release of extreme precipitation’ how does Figure 2b show an earlier release of precipitation? By how many hours is this release earlier?”

The change in storm timing is discussed in response to comment 13 by Reviewer #2. New analyses show that there is no clear change in timing of extreme thunderstorms.

Comment 25 (SI): “L28 I suggest redefining the difference between COSMO-CLM and COSMO-CLM2 here as I failed to notice the 2, which confused me. This difference is defined in the Methods section – this highlights the difficulty of having the information spread over three sources. Adding a table of model experiments would help.”

A table describing all model experiments is now included in the Supplementary information (see Comment 5 by Reviewer #2).

- P2L29 (SI): “In COSMO-CLM², the default land surface model of COSMO-CLM (TERRA-ML) is replaced by the Community Land Model version 3.5 (CLM3.5) (Davin et al., 2011; Davin and Seneviratne, 2012).”

Comment 26 (SI): “L68 ‘results are in close agreement’ I disagree with this statement. These plots are on a log scale, which minimises the differences to the reader. I think it would be better to say that the COSMO-CLM2 results are closer to the observations than CMIP5 or ERAI precipitation is.”

As the comparison between CORDEX or ERAI is already explained in the next phrases, we propose the following change:

- P3L58 (SI): “Results reveal a ~~close~~ **reasonable** agreement ...”

Comment 27 (SI): “L79 ‘propagation in westerly direction along the synoptic flow’ do you mean the storms are advected at the same speed as the synoptic flow or that the storms propagate faster than the synoptic flow due to the cold pool/regeneration of the convective lifecycle?”

By this we mean that storms are advected at the same speed as the synoptic flow:

- P3L79: “followed by intensification and ~~propagation~~ **advection** in westerly direction along the synoptic flow”

Comment 28 (SI): “L118-129 This text should probably come under a separate heading, as it doesn’t really address ‘assessing uncertainty’”

This paragraph was moved to a new subsection called ‘Correlation versus causality’. The second paragraph was modified and moved to the main text (see Comment 3 by reviewer #1).

Comment 29 (SI): “L136 What are the CORDEX models driven by? I don’t **think** this information is anywhere in the paper, though it may be hiding somewhere. If all the CORDEX models are driven by the same GCM it is not a surprise that they behave in a similar way (e.g. Fig S8).”

The 9 CORDEX simulations are driven by 9 different GCMs (see old Fig. S8). However, since we only included RCMs which have a lake model included, we ended up with only one RCM

(RCA4). This certainly influences the spread of the ensemble, however not including a lake model leads to unacceptable biases in terms of lake surface temperature (and thus mesoscale circulation; Thiery et al., 2015). All simulations, including their lateral boundary conditions, are now clearly described in Supplementary Table S1.

Comment 30 (SI): “L1146-175 Here you are assessing 99.9th percentile changes, which is one event in three years. The COSMO-CLM simulations are only 29 years long, so this translates to about 9 or 10 events. Is this enough to get statistical significance?”

Indeed, since the HIS and FUT simulation are only 30 years long, we expect the sample size of very extreme precipitation events (99.9th percentile) to contain only about 10 values. This is why the Clausius-Clapeyron scaling subsection is mostly focusing on the CORDEX ensemble. Here 150 years are available for each of the 9 members, hence we expect a total sample size of about 2 x 490 events (Lake Victoria and surrounding land, respectively). This should be sufficient to derive reliable Clausius-Clapeyron scaling factors. Although the sample size of HIS and FUT are much smaller, the distinctly stronger scaling factors over Lake Victoria are also found both in CORDEX and COSMO-CLM².

Comment 31 (SI): “L155 say upfront which models you are doing this analysis for.”

- **P5L155 (SI):** “we compute the 99.9th percentile precipitation per model **CORDEX simulation** and per decade during 1950-2100.”

Comment 32 (SI): “L169 why aren’t the COSMO-CLM results for the C-C scaling plotted on Figure S10?”

This is because the sample size of HIS and FUT for very strong precipitation events is very small (see also our response to Comment 30). The simulations count only 2 x 3 decades, which would result in only six points per geographic location and unreliable error bars.

Comment 33 (SI): “L176 By ‘moderate extreme’ do you mean the 99th percentile? By ‘distinct scaling pattern’ do you mean the land-lake difference?”

- **P6L176:** The distinct scaling ~~pattern over lake and land~~ for ~~moderate~~ very extreme precipitation is remarkable,...

Comment 34 (SI): “Figure S4 if the ERA-Interim data has been remapped onto the COSMO-CLM grid why are the grid sizes difference in Figure S4e-h compared with Figure S4i-l?”

This is because nearest neighbour interpolation was used to remap ERA-Interim to the COSMO-CLM² grid (see caption of Figure S4). With this interpolation technique the grid is refined without smoothing the image. This interpolation technique was chosen because the analysis focuses on extremes; it avoids smeared out extremes in space.

Reviewer #3

“This study has investigated the changing patterns of intense/extreme precipitation over Lake Victoria Basin (LVB) in equatorial East Africa. LVB is one of the regions which records the highest number thunderstorm days. The study reveals that there is significant intensification of extreme precipitation, consistent in the projections by CORDEX and COSMO-CLM models while there is significant decrease in the mean precipitation over LVB. The authors have used very robust methods of analysis while taking advantage of both model and satellite data. The discussions of the results are quite comprehensive and conclusions consistent with the analysis and model results. I have made the following specific comments and suggestions for improvement.”

We thank reviewer #3 for his/her evaluation of the manuscript. Below we address the issues that were raised.

Comment 1: “The driving mechanisms for the intensification of extreme precipitation over the Lake is attributed to increasing thunderstorm activities which are preceded by intense rainfall and storms over land. However, I find this to be somehow inconsistent with previous studies that reveal that over land areas in the entire equatorial eastern Africa region, rainfall has been consistently decreasing over the past decades unlike what the models (including majority of IPCC AR5) projections actually indicate. The trend currently referred to in some studies as the East Africa climate paradox (e.g. Rowell et al., 2015). Therefore the two theories proposed as the primary mechanisms through which increasing storms over land intensify the thunderstorms over the lake (e.g. page. 3, lines 93-97) are not fully supported by the observed decrease in rainfall over land. Based on large scale dynamics of the East Africa (LVB) rainfall (e.g. Nicholson et al. 2000) the decreasing rainfall trends over the region means that there is suppressed large scale horizontal moisture transport and convergence over land areas (may be based on decadal or longer variability).”

We thank the reviewer for raising this interesting point, which is food for reflection. We used this feedback to frame our research better in the existing knowledge.

First, It is important to distinguish clearly between *mean* and *extreme* precipitation changes over Lake Victoria, and this for three reasons:

- (i) **Different processes control their occurrence in the present-day climate:**
 - *Mean:* Lake evaporation is a key component of the hydrological cycle, supplying moisture. The lake/land breeze system redistributes the moisture and generates thunderstorms (see also our response to Comment 2 by reviewer #3).

- *Extremes*: mesoscale atmospheric dynamics are of key importance, evaporation much less. A thunderstorm is not going to be caused by more evaporation (Lake Victoria evaporates more during the dry season like every African Great Lake!). The explanations on P3L93-98 refer to this point (not to the future change).

(ii) **Their future change is of different sign (Illustration 9):**

- *Mean*: precipitation is expected to decrease over Lake Victoria. Over the surrounding land precipitation is projected to increase (CORDEX ensemble mean) or
- *Extremes*: extremes are projected to increase over Lake Victoria, and this more stronger than over surrounding land.

(iii) **Different processes control their future change:**

- *Mean*: precipitation changes are induced by changing mesoscale dynamics (land heating up faster compared to land)
- *Extremes*: precipitation changes are induced by changes in moisture advection. The converging air masses contain more moisture under future 'extreme' conditions.

Illustration 9 : a) Projected change of the precipitation distribution over Lake Victoria for wet days only (precipitation > 1 mm/day). b) zoom the red rectangle showing the change of the tail of the distribution.

In the conclusions of the revised manuscript, we now emphasize that a general drying over Lake Victoria is not in contradiction with an increase in extremes:

- P4L145: “In contrast to the average decrease, the rise in precipitation extremes is entirely due to enhanced future moisture availability (Fig. 4). Only over the lake the advection of more humid air supplies enough moisture to sustain Clausius-Clapeyron scaling. **The increase in extremes is therefore is not physically incongruous with the decrease in mean precipitation caused by mesoscale dynamical changes.**”

Second, we acknowledge that decadal variability in circulation and large-scale moisture transport may of course influence mean and extreme precipitation over the Lake Victoria basin. Smith et al. (2014) have, in fact, shown that these are important factors controlling lake level variations over Lake Victoria. However, in this study we wish not to focus on decadal precipitation variability. In the revised manuscript, we clearly identify the spatial and temporal scope of our study. By improving Fig. 4 of the paper (see response to comment 4 by reviewer #2) and making the following changes in the manuscript, we now convey our central message more clearly:

- **P5L160:** “High-resolution projections accounting for lake-atmosphere interactions are still very rare and may face challenges (Xie et al., 2015; Rowell et al., 2015), but adopting this approach is critical to assess future climate impacts in regions where lakes are abundant.”
- **P6L188:** “Overall, the simulations are designed to simulate the influence of a high-emission scenario on mean and extreme precipitation over and around Lake Victoria. Large-scale precipitation changes (e.g. over the whole of East Africa; Rowell et al., 2015) and influences of decadal variability (Nicholson, 2000; Smith et al., 2014) are thereby beyond the scope of this study.”

Third, we focus on the changes in precipitation over Lake Victoria, not over the whole of East Africa. In a study by Souverijns, Thiery, Demuzere and van Lipzig (Environmental Research Letters, in review), we find that future changes in *mean* precipitation over Lake Victoria are very different from the rest of East Africa (drying versus average wetting; Illustration 10). Hence we believe that it is possible to investigate precipitation changes over Lake Victoria without having to address changes over the whole of East Africa. We conclude that a discussion of the East African Climate paradox is beyond the (spatial) scope of this work.

Illustration 10 : Absolute precipitation changes from between 1981-2010 and 2071-2100 (ensemble mean of 15 CORDEX members under RCP8.5). Statistical significance at the 1 % level was tested by

applying the two-sided non-parametric Kolmogorov-Smirnov test on the annual mean precipitation amounts of the different members in the historical and future period and is denoted by the black diamonds. Adapted from Souverijns et al. (in review).

Comment 2: “The second inconsistency which is related to the above is the fact that the results of this study also shows that over-lake evaporation (precipitation recycling!) is also not likely to contribute the intensification of thunderstorms over the lake. I also find this as inconsistent given that many studies have consistently shown that the water balance over the lake is dominated by the near balance between ET and Precipitation. Could the models be misrepresenting the contribution of evaporation? If not, despite the sporadic intense storms over land areas that the study show to be the precursor of the intense over-lake thunderstorms, how would the intense thunderstorms be sustained as the land areas are getting drier? This is somehow alluded to on page 4 (lines 113-116), but for projections which also indicate reduction in moisture convergence by ~3%. What are the large-scale moisture transport mechanisms?”

Indeed many studies have shown that the water balance of Lake Victoria is dominated by the near-balance of precipitation and evaporation (see e.g. Smith et al., 2014 for an overview). However, this does not imply that water evaporated from the lake is directly recycled into precipitation. Many studies (e.g. Song et al., 2004; Anyah et al., 2006; Argent et al., 2014; Thiery et al., 2015) have highlighted the importance of the lake-land breeze system over Lake Victoria, and Thiery et al., 2015 showed that ET over the lake is nearly constant throughout the whole day. Evaporated moisture is thus transported away from the lake along with the lake breeze during daytime, whereas at night the land breeze advects moisture to the lake.

The proposed framework to explain climate change influences on mean and extreme precipitation is therefore not physically inconsistent with published water balance considerations.

- P2L46: “Local evaporation and mesoscale circulation have been identified as key drivers of the present-day diurnal cycle of precipitation over Lake Victoria (Song et al., 2004; Anyah et al., 2006a; **Argent et al., 2014; Chamberlain et al., 2014; Williams et al., 2014**; Thiery et al., 2015), but so far it is not known how mean and extreme precipitation over this lake respond to a temperature increase induced by anthropogenic greenhouse gas emissions.”

Observational estimates of evaporation over Lake Victoria are, to our knowledge, not available, and it is therefore very difficult to evaluate any model’s ability to represent evaporation directly. However, an indirect evaluation is possible. Thiery et al. (2015) showed that lake surface temperatures over Lake Victoria (and other African Great Lakes) are closely

reproduced, including spatial patterns within and between lakes (Illustration 11)². This is an indirect indication that the SEB components are reasonably well reproduced by the surface flux routines coupling the FLake to the atmospheric module in COSMO-CLM².

- P2L51 (SI):** “The mean annual cycles of net shortwave and longwave radiation at the surface, sensible and latent heat flux and cloud cover were also simulated mostly within the margins of observational uncertainty. **A direct evaluation of the sensible and latent heat flux over the lake surfaces is so far not possible due to the lack of reliable observational reference, but the close reproduction of the lake surface temperature patterns suggest that the surface energy balance is reasonably well reproduced also over the African Great Lakes.**”

Illustration 11: 1999-2008 observed lake surface water temperatures LSWT [K] from the ARC-Lake dataset (top panels) and modeled LSWT from the COSMO-CLM² CTL simulation (central panels) for (a,e) Lake Victoria, (b,f) Lake Tanganyika, (c,g) Lake Kivu and (d,h) Lake Albert. (i-l) Lake-averaged observed (black line) and modeled (green line) monthly mean LSWT including observational error estimate as provided with the product (red shading). Adapted from Thiery et al., 2015 (© American Meteorological Society).

² A warm bias of +.04°C remains over Lake Victoria, and the annual lake surface temperature cycle is generally overestimated (due to an underestimation of the mixed layer depth by the lake model).

Regarding the possible drying over land, we note that on average precipitation is projected to remain unchanged over the land surrounding Lake Victoria in COMOS-CLM² (Figure S5f), whereas in CORDEX precipitation even increases over the land surrounding the lake (Illustration 10). It is therefore not projected that the land areas are getting drier towards the future. And even if it were to get drier, an average drying over land would not preclude the occurrence of days with extreme land precipitation in the future. By analogy, the fact that mesoscale circulation is on average acting to decrease moisture convergence does not preclude the occurrence of days with atmospheric convergence as strong as it is today. Future projections indicate that such atmospheric convergence leads to even more moisture convergence than today as the air masses are more moist.

To make these points clear in the manuscript, we have included a new paragraph in the supplementary information describing the results of the Moisture convergence calculations (Fig. S11; see also comment 17 by reviewer #2):

- **P5L144 (SI): “Moisture Flux Convergence. By applying the Moisture Flux Convergence framework (MFC) to the COSMO-CLM² output, it is possible to quantify the moisture convergence over Lake Victoria for a given time period (Methods). Supplementary Fig. S8a shows the 24h average moisture fluxes in the CTL simulation. Spatial integration yields an average MFC of $258 \times 10^9 \text{ kg d}^{-1}$ from 1999 to 2008. While the atmospheric column loses moisture towards the west, the moisture gain from the east is larger, resulting in a net convergence of moisture. During 24h periods associated with extreme nighttime precipitation (Supplementary Fig. S8b), MFC more than triples to $807 \times 10^9 \text{ kg d}^{-1}$. The total change (Supplementary Fig. S8c) can subsequently be split up into a dynamic (Supplementary Fig. S8d) and thermodynamic contribution (Supplementary Fig. S8d; see Methods how this is done). Results show that the dynamic component is the dominant contributor (74%) to the increase in MFC during extremes (Supplementary Table S2). Note that the relative contribution to the precipitation change may depend on the considered percentile. Investigating this sensitivity is however beyond the scope of the present study. The same procedure was applied to quantify the future change in MFC during extremes only (Supplementary Table S2). In this case, the increase in MFC by +27% is entirely due to the thermodynamic component, that is, a higher moisture content of advected air masses instead of mesoscale circulation changes.”**

Comment 3: “On page 4 (lines 133-145) While the model projects general decrease in mean precipitation, again it is not clear what are the primary sources of moisture that triggers and sustains extreme thunderstorms over the lake.”

The primary sources of moisture for extreme thunderstorms over the lake is advection from over land through mesoscale circulation. Where this moisture eventually comes from (local

evaporation from the land around Lake Victoria, or being advected through the region along the synoptic flow) is beyond the scope of the present study, in fact this cannot be assessed with our high-resolution RCM framework. Moisture convergence is high enough to sustain and even intensify extreme thunderstorms into the future. This is clear from our moisture flux convergence calculations (see Supplementary figure S8 and supplementary table 2 in the revised manuscript). Which show that MFC during extremes increases by 27% towards the end-of-the-century. This is now clearly explained in the supplementary information (see previous comment).

Comment 4: “One page 4 (line 143) the approximate number of people directly supported by Lake Victoria is 35million (not 3.5 million).”

We corrected this mistake. As described in Semazzi et al. (2011), around 30 million people live at the shores of Lake Victoria, and the lake provides employment to 3-4 million people (hence our confusion).

- P4L143: “~~3.5~~ 30”

Reviewer #4

“The authors investigate extreme precipitation in the current climate and its response to climate change over Lake Victoria. They find that extremes will intensify in this region, mainly due to thermodynamic effects (increased atmospheric humidity).

The topic of the paper and the results are interesting, I think that the work shown in this manuscript can be published. But I think that significant clarifications and further investigations are needed to complete the study. I therefore cannot recommend publication in the current form.”

We thank the reviewer for the useful suggestions. Below we address the issues that were raised in the review.

Comment 1: “First my major overall comment is that correlation does not imply causality. Throughout the paper, conclusions are drawn about the causality between 2 variables based on correlation between the 2. I didn't find the arguments always convincing. For example, the authors find a correlation between the afternoon land breeze and extreme lake precipitation (figure 3d). It is not clear to me how afternoon land breeze can impact nighttime precipitation extremes, what is the physical mechanism? Probably the key variable controlling the dynamic contribution is the temperature difference between land and lake. Of course this is related to the afternoon land breeze as well, but the real physical process and causality is between the land/lake temperature contrast and extreme lake precipitation.”

Thank you for raising this very important point. We agree that it is not possible to establish causality for every possible relationship described in the paper. Therefore we reformulated several statements in more careful wordings (see next paragraph). On the other hand, we made an effort to derive some causal effects by comparing the control run to a ‘no-lake’ simulation (see further on).

First, we agree that the lake-land temperature contrast is a key variable to understand the processes at play. Illustration 12 shows this relationship. The lake breeze is a direct consequence of this temperature contrast, and consequently the relationship in Fig. 3b is very similar to the temperature contrast curve in Illustration 12. Note also that the lake surface temperatures show no relationship to nighttime precipitation. Due to the thermal inertia of the water body there are much less short-term variations in this variable compared to land temperature. Hence all dynamic contributions are expected to come from variability in land surface temperatures: lower land temperatures *reduce* the *afternoon* lake-land temperature contrast and in turn reduce the lake breeze, while they *enhance* the *nighttime* temperature contrast and thereby the land breeze. We subsequently state that a weaker

lake breeze contributes to nighttime storm intensity by preventing moisture from being transported away from the lake, and that the stronger land breeze contributes to nighttime storm intensity by increasing moisture convergence and column instability. Unfortunately the causality of this last step cannot be fully established. Nevertheless it is plausible that higher over-lake water vapour increases thunderstorm development (Mazon and Pino, 2013), while our MFC analysis clearly shows that moisture convergence more than triples during 24h periods associated with nighttime extreme precipitation.

Illustration 12 : afternoon lake-land temperature contrast versus nighttime lake precipitation. Binning procedures is the same as in Fig. 3 and the result is very similar to the lake breeze relationship.

Based on these considerations we decided to replace panel d of Fig. 3 by Illustration 12. We also phrase our hypothesis more cautiously now.

- P3L93: “These severe land storms **could** impact storm intensity over the lake in two ways. First, they enhance moisture convergence by increasing the near-surface specific humidity (thermodynamic control; Fig. 3c and Supplementary Fig. S11). Second, they modify the lake/land breeze system (Thiery et al., 2015) by cooling the land surface (dynamic control). **In that case** the cold pools of the afternoon storms **act to** reduce gradients in near-surface air temperature between lake and land (**Fig. 3d**), thereby weakening the lake breeze and **possibly also** moisture transport away from the lake (**Fig. 3d**). ~~Moreover, persistence of~~ **If** the cold anomaly **persists** into the night, **this could** strengthens the land breeze and by that **possibly** stimulates moisture convergence and column instability (**Mazon and Pino, 2013**).”

Second, of course there could be a mechanism that both increases the land breeze *and* the extremes. We have addressed that question by including an additional analysis, detailed hereafter.

One possible way to test causality in a modelling framework is to isolate a given effect using conceptual experiments. We have done this exercise by conducting a COSMO-CLM² simulation identical to CTL, except that we removed all lakes and replaced them by representative land (NOL, see Supplementary Table 2). If the observed relationships (Fig. 3b-d) in NOL would be identical to CTL, this would mean that the lake has no influence on storm activity in the region, and that the relationship is purely driven by large-scale processes such as moisture transport. If, on the other hand, the relationship would be completely absent in NOL, then the lake would be the sole cause of the obtained correlations.

As explained in the Supplementary Information, the results of this exercise indicate that a relationship between, for instance, afternoon land storm and nighttime lake storms exists in NOL, but that the relationship is much more pronounced in CTL. In particular, the mesoscale lake effect attributes for 74% of the median nighttime lake precipitation on average in each bin and for 43% in the bins above the 90th percentile and 43% in the bins above the 99th percentile (Illustration 13). Therefore we conclude that, while large-scale processes are an important driver of lake storms, the afternoon land storms act as a positive feedback for storm activity.

Illustration 13 : Large-scale versus mesoscale influence on the occurrence of nighttime precipitation over Lake Victoria.

In addition, it is possible that future changes in large-scale circulation and moisture transport will modulate the mesoscale interactions identified in the present study as an important component for understanding local climate change. Given the limited domain of our high-resolution model, this question cannot be addressed. Further research is required to fully understand large scale drivers of local precipitation variability and change in the Lake Victoria basin. This could, for instance, be achieved through surrogate climate scenarios

whereby thermodynamic changes play but atmospheric dynamics are not affected (e.g. Schär et al., 1996; van Lipzig et al., 2002).

We therefore made the following changes in the manuscript:

- P1L18: “Land precipitation on the previous day exerts a ~~strong~~ control on nighttime occurrence of extremes on the lake by enhancing atmospheric convergence (74%) and moisture availability (26%).”
- P3L93: “**Large-scale moisture availability contributes to this positive relationship, but alone it cannot explain the observed correlation (Supplementary Information). Land storms therefore act as a positive feedback for the intensity of nighttime lake storms.**”
- P3L109: “**A large fraction (74%)** of this increase can be attributed to dynamical effects, while only 26% is due to the enhanced moisture **content of converging air masses** ~~supplied by afternoon land precipitation~~ (Supplementary Fig. S8 and Table S2).”
- P4L117 (SI): “**Correlation versus causality.** Application of the binning procedure to observations and the CTL simulation highlights a strong relationship between afternoon land precipitation and nighttime lake precipitation over Lake Victoria (Fig. 3). By itself this relationship does not imply causality, in fact, there may be a third factor controlling both land and lake precipitation (e.g. synoptic-scale **moisture transport persistence**). Comparison of the CTL simulation to a simulation whereby all lake pixels have been replaced by representative land (NOL) allows us to separate the influence of Lake Victoria from all other effects, including synoptic-scale **moisture transport persistence** (Thiery et al., 2015). Results of this comparison indicate that the mesoscale lake effect attributes for ~~more than 80%~~ **74%** of the median nighttime lake precipitation in each bin, and for ~~48%~~ **43%** in the bins above the 90th percentile **and 34% in the bins above the 99th percentile** (~~nighttime NOL precipitation was binned according to nighttime lake precipitation from CTL~~). ~~From this analysis we conclude that mesoscale lake effects are the dominant cause of the observed positive relationship, although large-scale processes also contribute to it.~~ **From this analysis we conclude that large-scale processes alone cannot explain the occurrence of extremes over Lake Victoria, and that mesoscale lake effects are a positive feedback enhancing nighttime thunderstorm intensity.** Our results also highlighted the importance of changes in local atmospheric moisture content and mesoscale processes for understanding the future intensification of precipitation extremes over Lake Victoria. However, it is possible that large-scale variability and climate change modulate these mesoscale interactions. Given the limited domain of our high-resolution model, this question cannot be fully addressed in the present study. Further research is required to fully understand large scale drivers of local precipitation variability and change in the Lake Victoria basin. This could, for instance, be achieved through surrogate climate scenarios whereby thermodynamic changes play but atmospheric dynamics are not affected (e.g. Schär et al., 1996; van Lipzig et al., 2002).”

Comment 2: “Another remark about figure 3d is that the extreme precipitation actually appears to be largely insensitive to the lake breeze (flat curve) except the last decade of precipitation. What precipitation percentile does that correspond to?”

All curves in Fig. 3 are showing ‘scatter plots’ of binned data. For instance, Fig. 3c is showing the median and 25-75 percentile range of the 100 afternoon land specific humidity bins versus the average nighttime lake precipitation of that bin (not versus time). Nighttime precipitation was used to bin afternoon specific humidity on land with a 1% bin width. The last 10 points on the right hand side thus corresponding to the nights with precipitation above the 90th percentile.

The flat curve in Fig. 3d therefore indicates that “non-extreme” nighttime precipitation is not sensitive to lake breeze strength; towards the higher percentiles there appears to be a strong relationship (extreme precipitation events are associated with weak lake breezes). This plot is now better explained in the manuscript:

- **Caption fig. 2: “Afternoon controls on nighttime extreme precipitation. a,** Afternoon SEVIRI OT detections over land surrounding Lake Victoria ~~binned according to~~ **versus** nighttime OTs over the lake (2005-2013; blue). **b,** Afternoon TRMM 3B42 precipitation around Lake Victoria ~~binned according to~~ **versus** nighttime precipitation over the lake (1998-2013; red) and corresponding modelled values from a 10-year reanalysis downscaling with COSMO-CLM² (1999-2008; brown) (Methods). **c-d,** Same as **b,** but for the afternoon land 2-meter specific humidity ($Q_{V,2M}$) and lake breeze strength, respectively, as derived from the reanalysis downscaling. The lake breeze strength is computed here as the average 10 m wind speed radiant from a simplified Lake Victoria contour (Supplementary Fig. S1). **Each variable on the y-axis was binned according to the variable on the x-axis using a bin width of 1%.** Full lines indicate the bin median and uncertainty bands the interquartile range. Note the logarithmic x-axis.”

Comment 3: “Another example is the increased moisture interpreted as being the result of enhanced land storms. I am not a specialist of this region, but I think one could argue the opposite causality: increased land storms being the result of increased moisture, which could be due to another source (large scale advection...). I don't think that the analysis of this manuscript is sufficient to deduce that the afternoon land storms are responsible for, and not caused by, the enhanced moisture. There may be other moisture sources, and both land and lake OT high counts would result from enhanced moisture.”

This comment has been addressed in response to comment 1 by reviewer #4.

Comment 4: “Also I found the presentation and wording not always clear. For example:- Figure 2 the color bar convention switches between panels a and b.”

We decided to adhere different color bars to panels a and b because they display different quantities (Extreme precipitation and extreme precipitation change). However, we acknowledge that it is not straightforward to know what is plotted without reading the caption. We have therefore updated the figure (color bar labels) to better indicate what is plotted.

Comment 5: “- Regarding the OT counts (line 46, Methods) from the methods I understand that adjacent OT pixels are identified as being part of the same OT. But line 46 mentions 1,400,000 OTs over the lake alone, for the 9 years investigated that yields more than 400 OTs per 24h. I guess this means that several OTs are found in one thunderstorm, I think that the definition and physical meaning of OT could be clarified (at least in Methods).”

This is a correct remark. Our dataset displays the number of OT “pixels” that were detected over the region. This is a metric of the areal coverage of OT regions, not necessarily the number of “storms”. An OT region can be composed of several adjacent pixels, typically not exceeding 15 km in diameter (i.e. maximum 4x4 or 16 SEVIRI pixels). The data is provided this way because it facilitates assigning OT to a grid (spatial patterns are much more coherent than if we would select only one pixel per OT).

In addition, we analysed 1 year of OT data over the African Great Lakes (see Supplementary Fig. 1 for domain) and found that the mean number of pixels per OT was 11 (with a standard deviation of 3 pixels). 1 year of data therefore has over 130 000 OTs for the African Great Lakes region.

- P2L46: “~~OTs~~ OT pixels.”
- P2L47: “~~OTs~~ OT pixels.”
- P2L65: “Using this approach more than 50 million OT pixels were detected from 2005-2013 over equatorial East Africa. **A single OT is on average composed of 11 OT pixels and typically does not exceed 15 km in diameter.**”
- Fig. 1: “OT pixel counts.”
- Caption fig. 3: “Afternoon SEVIRI OT pixel detections over land surrounding Lake Victoria versus nighttime ~~OTs~~ OT pixels over the lake”

Comment 6: “- line 62-66: "the spatial pattern indicates an eastward shift... in the future storms develop faster leading to more extreme precipitation in the east" This is speculative, I do not think that the distribution of figure 2 allows you to reach this conclusion. Have the authors quantified the time duration of those thunderstorms, to check whether it is shorter with warming? Alternatively, the speed of the trade winds may change (leading to different speed of advection over the lake). Or the storms could be initiated further east compared to current climate.”

This is a very valid point. Figure 2b shows a spatial shift but this does not imply a temporal change. We therefore investigate a possible change in storm timing by looking at the diurnal cycle of precipitation in present and future climate conditions, and this both for mean and extremes (Illustration 14).

Illustration 14 : projected future change in the diurnal cycle of (a) mean and (b) extreme precipitation over Lake Victoria.

Results indicate that the diurnal cycle of both mean and extreme precipitation are not changing towards the future. We updated the manuscript to be consistent with these new results:

- ~~P2L65: “The spatial pattern of change indicates an eastward shift of intense precipitation systems (Fig. 2a-b). Today convection initiates in the eastern third part of the lake and intensifies while being advected westwards along the trade winds (Carpenter, 1922; Thiery et al., 2015). In the future, storms are projected to release to develop faster leading to more extreme precipitation more in the eastern part of the lake, leading to an eastward shift of intense precipitation (Fig. 2a-b).”~~

Comment 7: “- line 109: "enhanced moisture supplied by afternoon land

precipitation" How do you know that the moisture is supplied by land precipitation? There could be other moisture sources (in particular large-scale advection)."

Indeed we rephrase this more accurately now:

- P3L109: "74% of this increase can be attributed to dynamical effects, while only 26% is due to the enhanced moisture **content of converging air masses** supplied by afternoon land precipitation (Supplementary Fig. S8 and Table S2)."

Comment 8: "- Figure 2: are all the panels a-d from the COSMO-CLM simulations? "The clear dipole pattern of change indicates an earlier release of extreme precipitation": I did not find this wording very clear."

In Figure 2, Panels a-c are for COSMO-CLM², panel d is for CORDEX. We removed the sentence referring to the dipole pattern since new analyses showed that there is no change in extreme storm timing. Hence the new figure caption:

- Caption Fig. 2: "Projected end-of-century changes in extreme precipitation over Lake Victoria. **a**, Nighttime 99th percentile precipitation ($P_{99\%,\text{night}}$, 00h-09h UTC) and **b**, its projected future change according to **from** the high-resolution COSMO-CLM² model. ~~The clear dipole pattern of change indicates an earlier release of extreme precipitation.~~ **c-d**, 24h Lake (blue bars) and surrounding land (red bars) binned precipitation change from COSMO-CLM² and the ensemble mean of nine CORDEX-Africa members, **respectively**. The red rectangle in Supplementary Fig. S1 includes the land pixels considered as "surrounding land". All changes are between time periods 1981-2010 and 2071-2100 under RCP8.5."

Comment 9: "- line 108 and fig3c-d: it looks like the dynamic control dominates only the last decade, before that the thermodynamic effect is larger. So which effect dominates may depend on the precipitation percentile considered."

This is an absolutely plausible hypothesis, and the relative contribution for other percentiles than the upper one could also be quantified using the MFC framework (Fig. S8 of revised paper). Yet in this study we are precisely trying to explain the occurrence of the extreme precipitation percentiles. In that sense we are most interested in the upper decile (or even upper percentile) of the nighttime precipitation distribution (so the points towards the right). We quantified the contribution of both effects for the upper percentile using the MFC

framework. Computing this contribution for every nighttime precipitation percentile is certainly possible, but beyond the scope of the present study.

- P5L144 (SI): **“Note that the relative contribution to the precipitation change may depend on the considered percentile. Investigating this sensitivity is however beyond the scope of the present study.”**

References

- Anyah, R. O., Semazzi, F. H. M. & Xie, L. Simulated Physical Mechanisms Associated with Climate Variability over Lake Victoria Basin in East Africa. *Monthly Weather Review* 134, 3588-3609 (2006).
- Argent, R., Sun, X., Semazzi, F. H. M., Xie, L. & Liu, B. The Development of a Customization Framework for the WRF Model over the Lake Victoria Basin, Eastern Africa on Seasonal Timescales. *Advances in Meteorology* 653473 (2014).
- Ban, N., Schmidli, J. & Schaer, C. Evaluation of the convection-resolving regional climate modeling approach in decade-long simulations. *Journal of Geophysical Research: Atmospheres* 119, 7889-7907 (2014).
- Ban, N., Schmidli, J. & Schaer, C. Heavy precipitation in a changing climate : Does short term summer precipitation increase faster? *Geophysical Research Letters* 42, 1165-1172 (2015).
- Bedka, K. Overshooting cloud top detections using MSG SEVIRI Infrared brightness temperatures and their relationship to severe weather over Europe? *Atmospheric research* 99, 175-189 (2011).
- Brisson, E. et al. How well can a convection-permitting climate model reproduce decadal statistics of precipitation, temperature and cloud characteristics? *Climate Dynamics* (2016).
- Chamberlain, J. M. et al. Forecasting storms over Lake Victoria using a high resolution model. *Meteorological Applications* 21, 419-430 (2014).
- Chan, S. C., Kendon, E. J., Roberts, N. M., Fowler, H. J. & Blenkinsop, S. Downturn in scaling of UK extreme rainfall with temperature for future hottest days. *Nature Geoscience* 5, 1-6 (2015).
- Descy, J.-P. et al. East African Great Lake Ecosystem Sensitivity to changes. Tech. Rep., Belgian Science Policy, Brussels, Belgium (2015).
- Docquier, D., Thiery, W., Lhermitte, S., van Lipzig, N. Multiyear wind dynamics around Lake Tanganyika. *Climate Dynamics* (2016).
- Kendon, E. J., Roberts, N. M., Senior, C. A. & Roberts, M. J. Realism of rainfall in a very high-resolution regional climate model. *Journal of Climate* 25, 5791-5806 (2012).
- Kendon, E. et al. Heavier summer downpours with climate change revealed by weather forecast resolution model. *Nature Climate Change* 4, 570-576 (2014).
- Lauwaet, D., van Lipzig, N.P.M., Van Weverberg, K., De Ridder, K., Goyens, C., The precipitation response to the desiccation of Lake Chad. *Quarterly Journal of the Royal Meteorological Society* 138, 707-719 (2012).
- Mazon, J., Pino, D. The role of sea-land air thermal difference, shape of the coastline and sea surface temperature in the nocturnal offshore convection. *Tellus A* 65, 1-13 (2013).
- Mazon, J., Pino, D. Mesoscale numerical simulations of heavy nocturnal rainbands associated with coastal fronts in the Mediterranean Basin. *Natural Hazards and earth system sciences* 14(5), 1185-1194 (2014).

- Mazon, J., Pino, D. Role of the nocturnal coastal-front depth on cloud formation and precipitation in the Mediterranean basin. *Atmospheric Research* 153, 145-154 (2015).
- Prein, A. F. et al. A review on regional convection-permitting climate modeling: Demonstrations, prospects, and challenges. *Reviews of Geophysics* 53, 323-361 (2015).
- Proud, S. R. Analysis of overshooting top detections by Meteosat Second Generation: A 5-year dataset. *Quarterly Journal of the Royal Meteorological Society* 141, 909-915 (2015).
- Fensholt, R. et al. Analysing the advantages of high temporal resolution geostationary msg seviri data compared to polar operational environmental satellite data for land surface monitoring in Africa. *International Journal of Applied Earth Observation and Geoinformation* 377 13, 721-729 (2011).
- Schär, C., Frei, C., Lüthi, D., Davies, H., Surrogate climate-change scenarios for regional climate models, *Geophysical Research Letters* 6, 669-672 (1996).
- Seneviratne, S. I., Donat, M. G., Pitman, A. J., Knutti, R. & Wilby, R. L. Allowable CO2 emissions based on regional and impact-related climate targets. *Nature* 529, 477-483 (2016).
- Souverijns, N., Thiery, W., Demuzere, M., van Lipzig, N. Drivers of future changes in East African precipitation, *Environmental Research Letters* (in review).
- Smith, K. A. & Semazzi, F. H. M. The Role of the Dominant Modes of Precipitation Variability over Eastern Africa in Modulating the Hydrology of Lake Victoria. *Advances in Meteorology* 2014, 1-11 (2014).
- Song, Y., Semazzi, F. H. M., Xie, L. & Ogallo, L. J. A coupled regional climate model for the Lake Victoria basin of East Africa. *International Journal of Climatology* 24, 57-75 (2004).
- Thiery, W. et al. Understanding the performance of the FLake model over two African Great Lakes. *Geoscientific Model Development* 7, 317-337 (2014a).
- Thiery, W. et al. LakeMIP Kivu: Evaluating the representation of a large, deep tropical lake by a set of 1-dimensional lake models. *Tellus A* 66, 21390 (2014b).
- Thiery, W. et al. The Impact of the African Great Lakes on the Regional Climate. *Journal of Climate* 28, 4061-4085 (2015).
- Tiedtke, M. A comprehensive mass ux scheme for cumulus parameterization in large-scale models. *Monthly Weather Review* (1989).
- Van Lipzig, N.P.M., van Meijgaard, E., Oerlemans, J., Temperature Sensitivity of the Antarctic Surface Mass Balance in a Regional Atmospheric Climate Model. *Journal of Climate* 19, 2758-2774 (2002).
- Williams, K., Chamberlain, J., Buontempo, C. & Bain, C. Regional climate model performance in the Lake Victoria basin. *Climate Dynamics* (2014).
- Zipser, E. J., Cecil, D. 317 J., Liu, C., Nesbitt, S.W. & Yorty, D. P. Where are the most intense thunderstorms on Earth? *Bulletin of the American Meteorological Society* 87, 1057-1071 (2006).

Reviewers' comments:

Reviewer #1 (Remarks to the Author):

The paper has been substantially improved by this revision and I am happy that it's now in a suitable form for publication.

It is disappointing, though, that the uncertainties have not been treated more thoroughly. I do understand the time-limitations of running large scale climate models but this should not be used as an excuse for presenting a less rigorous analysis.

The fact that it is a recurrent problem for high-res climate change studies says a lot about the current state of the field. However, as this appears to be a common problem then it is not feasible to hold the authors to blame for it, therefore I recommend publication of the manuscript.

Reviewer #2 (Remarks to the Author):

I was pleased to see a comprehensive response to the comments of myself and the other reviewers. The subsequent changes to the manuscript are a vast improvement compared to the originally submitted version. I can now recommend this manuscript for publication in its current form.

Reviewer #3 (Remarks to the Author):

As indicated in an email earlier, based on the authors' satisfactory responses to all my comments, concerns, and suggestions during my first review of the manuscript, I would like to recommend that the manuscript be considered for publication.